# SLAMF6 compartmentalization enhances T cell functions

Yevgeniya Gartshteyn[1,2], Anca D Askanase[1,2], Ruijiang Song[2], Shoiab Bukhari[2], Matthew Dragovich[2], Kieran Adam[2], Adam Mor[1,2,3]

Signaling lymphocyte activation molecule family member 6 (SLAMF6) is a T cell co-receptor. Previously, we showed that SLAMF6 clustering was required for T cell activation. To better understand the relationship between SLAMF6 location and function and to evaluate the role of SLAMF6 as a therapeutic target, we investigated how its compartmentalization on the cell surface affects T cell functions. We used biochemical and co-culture assays to show that T cell activity is enhanced when SLAMF6 colocalizes with the CD3 complex. Mechanistically, co-immunoprecipitation analysis revealed the SLAMF6-interacting proteins to be those essential for signaling downstream of T cell receptor, suggesting the two receptors share downstream signaling pathways. Bispecific anti-CD3/SLAMF6 antibodies, designed to promote SLAMF6 clustering with CD3, enhanced T cell activation. Meanwhile, anti-CD45/SLAMF6 antibodies inhibited SLAMF6 clustering with T cell receptor, likely because of the steric hindrance, but nevertheless enhanced T cell activation. We conclude that SLAMF6 bispecific antibodies have a role in modulating T cell responses, and future work will evaluate the therapeutic potential in tumor models.

## Introduction

Activation of T cells occurs when the CD3 T cell receptor (TCR) is engaged by a cognate peptide, the latter presented on APC. This interaction occurs at the immunologic synapse (IS) and is fine-tuned by co-receptors that colocalize with the CD3 in the contact zones of the IS. Co-receptor signals may be activating, as in the case of CD28, or inhibitory, as in the case of programmed cell death protein 1 (PD-1) and cytotoxic T lymphocyte–associated protein 4. Furthermore, the recruitment or exclusion of the co-receptors to and from the IS affects how these receptors modulate T cell functions.

Signaling lymphocyte activation molecule (SLAMF6) (*Ly108* in mice or *NTB-A* in humans) is a cell surface receptor expressed on a wide variety of hematopoietic cells. The homophilic interaction of

SLAMF6 in trans between T cells and APC promotes a stable cell–cell interaction. Activation of the SLAMF6 co-receptor on T cells results in recruitment of SRC family kinases and subsequent phosphorylation of cytoplasmic immunoreceptor tyrosine–based switch motifs. These secondary signals culminate in immunomodulation of the TCR response.

SLAMF6 activity affects T cell functions and can modulate the anticancer immune responses, suggesting potential for pharmaceutical targeting in cancer immunotherapy (1, 2). At the same time, the exact role of SLAMF6 in T cell signaling is not well understood, and contradictory reports of SLAMF6 enhancing or dampening T cell activity have been described (3, 4). In vitro antibody ligation of SLAMF6 cross-linked with activated CD3 results in increased T cell proliferation and IFN-γ release, suggesting that SLAMF6 is a co-stimulatory receptor (5). In addition, our group has previously shown that SLAMF6-mediated T cell activation is dependent on the clustering of SLAMF6 endodomain with CD3, highlighting that spatial proximity of the receptors and their downstream signaling molecules within the IS is necessary for T cell activation (6).

In this study, we investigated how compartmentalization of SLAMF6 on the cell surface affects its function. We performed a series of biochemical and co-culture experiments to either enhance or prevent SLAMF6 clustering with the TCR–CD3 complex, showing that these interventions resulted in an increase and decrease in T cell excitation, respectively. We then bioengineered bispecific antibodies to modulate SLAMF6 compartmentalization, showing that T cell activity can be regulated by targeting receptor colocalization on the cell plasma membrane.

## Results

### SLAMF6-mediated T cell activation requires spatial colocalization of SLAMF6 and CD3 along the cell membrane

In previous work, we had shown that SLAMF6 engagement, in the presence of CD3 ligation, results in increased T cell activity as measured by cytokine secretion (IL-2 and IFN-γ), cell proliferation, and phosphorylation of proximal TCR signaling proteins (6). This activation is dependent on colocalization of SLAMF6 with the CD3 in

---

[1]Division of Rheumatology, Department of Medicine, Columbia University Medical Center, New York, NY, USA    [2]Columbia Center for Translational Immunology, Columbia University Medical Center, New York, NY, USA    [3]Herbert Irving Comprehensive Cancer Center, Columbia University Medical Center, New York, NY, USA

Correspondence: yg2372@cumc.columbia.edu

the IS. To evaluate whether SLAMF6 signaling can be modulated by changing the spatial location of the receptor with respect to the TCR, we set up two stimulation conditions to mimic SLAMF6 receptor clustering versus separation from the CD3. In the first condition, T cells were stimulated with immobilized anti-CD3 and anti-SLAMF6 antibodies on a plate surface, thereby allowing the two receptors to cluster together. In the second condition, T cells were stimulated with immobilized anti-CD3 but soluble anti-SLAMF6 antibodies, introducing spatial separation between the activated SLAMF6 and CD3 receptors on the cell surface (Fig 1A). Equal amount of anti-SLAMF6 was ensured across all conditions.

Ligation of the SLAMF6 and CD3 receptors in the plate-bound SLAMF6 (plate) condition resulted in an increase in T cell proliferation as compared with ligation of CD3 alone (soluble) (Fig 1B). However, stimulation in the soluble SLAMF6 condition resulted in inhibition of T cell proliferation that occurred as a result of spatial separation of CD3 and SLAMF6 at the time of receptor engagement by anti-CD3 and anti-SLAMF6 antibodies, respectively. Cell surface expression of the activation markers CD25 (Fig 1C left panel) and PD-1 (Fig 1C right panel) and the levels of secreted IL-2 (Fig 1D) and IFN-y (Fig 1E) were decreased in the soluble SLAMF6 as compared with the plate-bound SLAMF6 condition, mirroring the decreased activation that was seen in the proliferation assay. The gating strategy is shown in Fig S1. Similarly, after a brief 4-h stimulation, there was a mild increase in intracellular IL-2 levels in the plate-bound SLAMF6 as compared with the soluble SLAMF6 condition (Fig 1F). We also evaluated T cell differentiation after a 5-d stimulation. Cell states were defined as naïve (N; CD45RA⁺CCR7⁺), central memory (CM; CD45RA⁻CCR7⁺), effector memory (EM; CD45RA⁻CCR7⁻), and terminally differentiated effector memory (TEMRA; CD45RA⁺CCR7⁻). In the plate-bound SLAMF6 condition, as compared with the soluble SLAMF6 condition, percentages of N cells were similar, CM were reduced, whereas those of EM and TEMRA cells were increased (Fig 1G). The mean T cell maturation indices ([1×N + 2×CM + 3×EM + 4×TEMRA]/4) for cells stimulated with anti-CD3 antibody alone, anti-CD3 + anti-SLAMF6 (plate), and anti-CD3 + anti-SLAMF6 (soluble) were 50.1, 59.4, and 51.4, respectively (one-way ANOVA $P$ = 0.001). Finally, we quantified the cell number of Jurkat T cells stimulated in culture over 3 d. The absolute cell count was significantly greater in the plate-bound SLAMF6 condition as compared with the soluble SLAMF6 stimulation condition (Fig 1H). Thus, ligation of SLAMF6 in proximity to CD3 results in net activation of T cell signaling, as measured by cell proliferation, up-regulation of cell surface activation markers, cytokine release, and T cell maturation. Spatially removing the SLAMF6 receptor from the CD3 receptor dampens this activation signal and constrains T cell proliferation. Hence, the location of SLAMF6 in reference to CD3 modulates the net effect of SLAMF6 signaling on T cell activity, with the potential to alter the signal from activation to inhibition.

### SLAMF6 co-immunoprecipitation (co-IP) assay identifies downstream proteins that bridge SLAMF6 and TCR signaling

Having observed that SLAMF6 has the ability to both excite and dampen T cell functions, modified by its location with respect to the CD3, we hypothesized that signal regulations are dependent on the availability of downstream proteins that bridge the SLAMF6 and CD3

signals. Specifically, activation after ligation of SLAMF6 and/or CD3 is dependent on SRC tyrosine kinases (LCK and FYN) that are brought into close proximity with their substrates when SLAMF6 and CD3 cluster together in the IS. To evaluate whether forced removal of SLAMF6 from CD3 results in a "steal" of signaling molecules away from the TCR, we performed a SLAMF6 co-IP assay with SLAMF6 clustered with CD3 (plate-bound SLAMF6) as compared with SLAMF6 that is spatially removed from CD3 (soluble SLAMF6). Jurkat T cells expressing V5-tagged SLAMF6 were stimulated with anti-CD3 and anti-SLAMF6 antibodies (plate SLAMF6 versus soluble SLAMF6 conditions, as above), after which cells were lysed and immuno-precipitated with anti-V5 antibodies to enrich the SLAMF6 interactome. The SLAMF6-immunoprecipitated reaction was submitted for mass spectrometry analysis to identify SLAMF6-interacting proteins. We hypothesized that key regulatory signaling proteins would remain associated with SLAMF6 in the soluble SLAMF6 condition, suggesting their removal from the CD3 site contributes to the inhibition seen in the soluble as compared with the plate SLAMF6 condition. In the analyzed co-IP samples, the bait, SLAMF6, was present in equal amounts in the plate and soluble SLAMF6 conditions, reflecting the quantitative robustness of both the co-IP conditions in addition to the quality control assessment in terms of equal loading of peptides from each condition.

An unbiased protein enrichment pathway analysis identified that the TCR signaling pathway and the primary immunodeficiency pathway, regardless of plate versus soluble SLAMF6 stimulation, are the most likely pathways to interact with SLAMF6 (Fig 2A). This intimate association of SLAMF6 pull-down proteins with TCR signaling proteins highlights why receptor clustering along the cell surface membrane is so important for downstream signal propagation. Subsequently, we restricted our analysis to proteins involved in the proximal T cell signaling pathway; these included LCK, FYN, and ZAP70 among others (Fig 2B). We found a non-differential pull-down of essential T cell signaling proteins in both the soluble and plate SLAMF6 conditions, suggesting they remain associated with SLAMF6 even when the latter is spatially removed from the CD3 site. Because SLAMF6, when spatially removed from CD3, was still associated with many of the signaling proteins essential for TCR signaling, we consider this "steal" of LCK, FYN, and ZAP70 away from the CD3 to be a contributing factor to the inhibition of TCR activity seen in the soluble SLAMF6 condition. A schematic diagram of the identified SLAMF6 interactome in TCR signaling is shown (Fig 2C). The tight network of association that exists between the proteins identified in the SLAMF6 interactome and those known to be essential for the proximal TCR signaling cascade is evident. Finally, we performed a protein–protein interaction analysis to identify the downstream kinases that are predicted to function downstream of SLAMF6 activation in both plate versus soluble SLAMF6 stimulation (Fig S2). The predicted kinases differ in the two stimulation conditions, suggesting that although both plate and soluble SLAMF6 stimulation signal via the TCR, these two activation conditions are not identical in downstream signaling events.

In summary, we took advantage of two T cell stimulation conditions, activation with plate-bound anti-SLAMF6 and partial inhibition with soluble anti-SLAMF6, to identify the intracellular binding partners associated with the SLAMF6 receptor. Proteins involved in T cell signaling pathways predominated in both

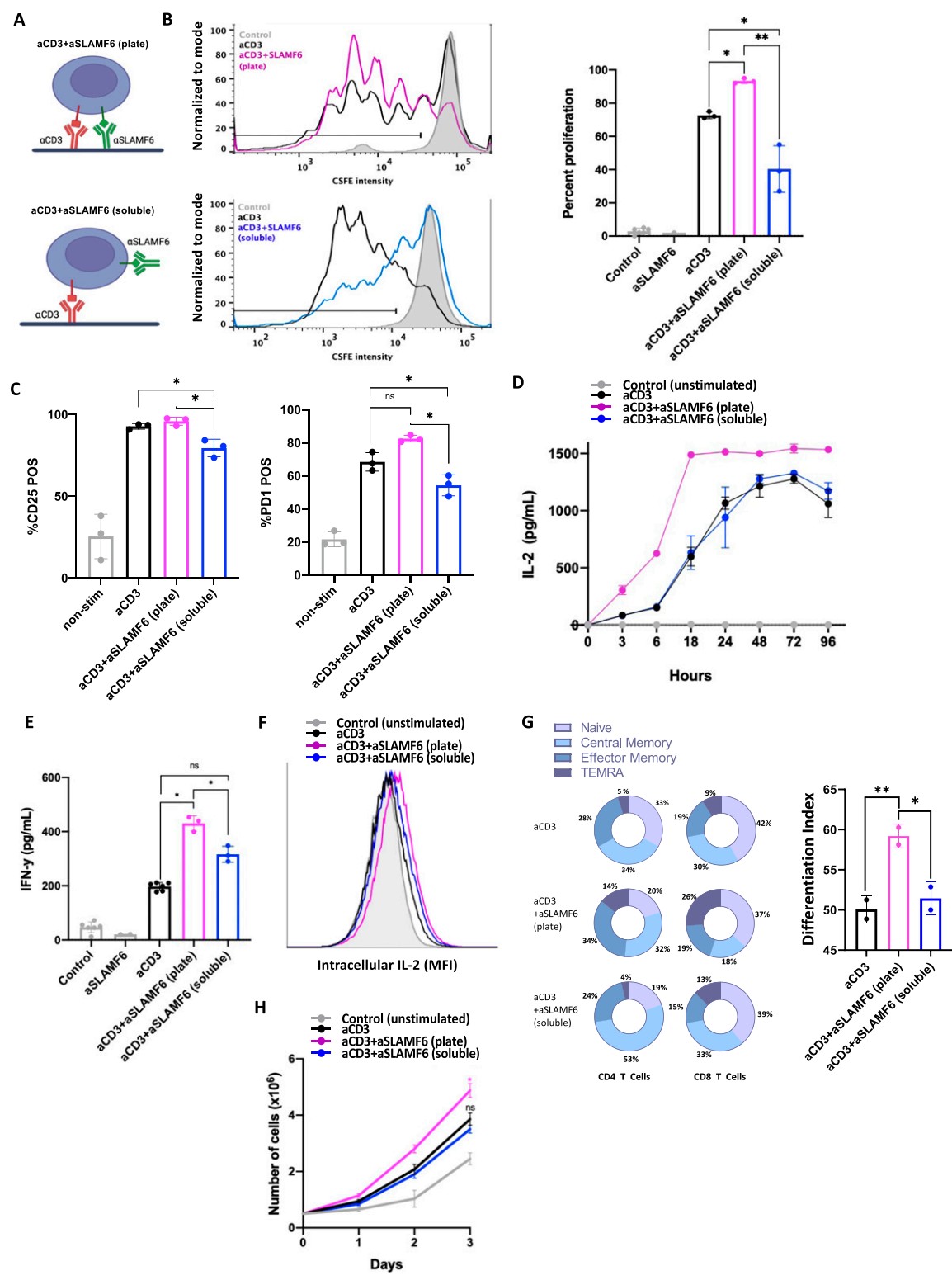

**Figure 1. Separation of SLAMF6 and CD3 inhibits T cell proliferation.**
**(A)** A schematic representation of the assays used to study SLAMF6 compartmentation. T cells were stimulated with either immobilized anti-CD3 (αCD3) and anti-SLAMF6 (αSLAMF6) on a plate surface (*top*) or immobilized αCD3 but soluble αSLAMF6 (*bottom*). **(B)** Freshly isolated primary CD3⁺ T cells were stained with CFSE then cultured in the presence of plate-coated αCD3 or plate-coated αCD3 + αSLAMF6 or plate-coated αCD3 + soluble αSLAMF6. After 120 h, the cells were assayed for FITC fluorescence for three independent experiments (n = 3). The data were analyzed for percent (%) of proliferating cells as depicted (black line); plate-coated αCD3 + αSLAMF6 resulted in greater proliferation as compared with αCD3 alone, whereas addition of soluble αSLAMF6 inhibited proliferation. **(C)** CD25 and PD-1 expression at 120 h was analyzed using flow cytometry. **(D, E)** Freshly isolated primary CD3 T cells were cultured in the presence of plate-coated αCD3 + αSLAMF6 or plate-coated αCD3 + soluble αSLAMF6. **(D, E)** The

conditions. We found no differential expression of proximal signaling proteins, suggesting that these SLAMF6-associated signaling molecules move along the cell surface together with the receptor. Thus, when SLAMF6 is spatially separated from the CD3 receptor, it has the potential to "steal" the downstream signaling molecules involved in TCR activation, thereby contributing to dampened cell activation.

### SLAMF6-mediated T cell activation is enhanced when SLAMF6 and CD3 cluster in the IS

We hypothesized that the function of the SLAMF6 receptor is dependent on its ability to translocate to the IS at the time of cell stimulation. First, we wanted to visualize the location of the SLAMF6 receptor with reference to the IS. Jurkat T cells were transfected with LifeAct mCherry (actin label) and GFP-SLAMF6, then co-cultured with Raji B cell loaded with Staphylococcus enterotoxin E (SEE) (Fig 3A). The IS was defined by visualization of actin accumulation at the contact zone. Microscopy imaging revealed that there were two predominant patterns of SLAMF6 expression. Specifically, in some cells, SLAMF6 expression was evenly distributed across the cell surface membrane (Fig 3B top row), whereas SLAMF6 enrichment at the site of the IS could be seen in others (Fig 3B bottom row). The former occurred in resting T cells and in 15% of the cells that formed synapse with another cell, whereas the remaining 85% of cells that formed synapses showed SLAMF6 enrichment at the site of the contact. Thus, we visually observed that SLAMF6 is able to translocate and cluster in the IS after cell stimulation.

To test our hypothesis that physical separation versus clustering of the SLAMF6 with the CD3 receptor affects T cell function, we designed beads conjugated with anti-CD3, anti-SLAMF6, or anti-CD3 + anti-SLAMF6 antibodies (Fig 3C). Compared with T cells stimulated with anti-CD3 alone, stimulation with anti-CD3 + anti-SLAMF6–conjugated beads resulted in increased IL-2 release, whereas stimulation with anti-CD3 beads admixed with anti-SLAMF6 beads failed to augment T cell activity with a trend toward inhibition as compared with stimulation with anti-CD3 alone (Fig 3C). We repeated the experiment using an Fc cross-linking antibody to cluster the anti-CD3 and anti-SLAMF6–stimulating antibodies on the surface of the T cell. We again saw increased IL-2 secretion as compared with stimulation with anti-CD3 alone or anti-CD3 and anti-SLAMF6 randomly admixed in solution (Fig 3D), suggesting that colocalization of the SLAMF6 with the CD3 augments downstream T cell activation.

We were thus able to show that SLAMF6 receptors can be found either dispersed from or clustered with the CD3 receptor in the IS. Physically promoting SLAMF6 clustering with CD3 at the time of stimulation, using either a system of conjugated beads or cross-linkers, resulted in enhanced T cell activation and established

SLAMF6 as an activating co-receptor when localized in proximity to CD3. On the other hand, T cell activation was dampened when SLAMF6 was spatially removed from the CD3 complex, suggesting loss of synergism between these signaling pathways.

### T cell stimulation with anti-CD3/SLAMF6–bispecific antibody enhances T cell activity

If the signal from a co-receptor can be further adjusted by changing its location along the cell surface membrane, we wondered whether interventions targeting receptor clustering may have a role in regulating cell signaling. We designed a bispecific monoclonal antibody simultaneously targeting CD3 and SLAMF6 (anti-CD3/SLAMF6) with the a priori hypothesis that it will augment T cell activation by promoting, at the time of receptor engagement, the clustering of SLAMF6 in proximity to the CD3 (Fig 4A). The bispecific antibody was generated by co-expression of monovalent OKT3-IgG-hole and monovalent anti-SLAMF6-IgG-knob constructs in 293 cells (Table S1). We validated the bispecific antibody binding by means of an ELISA assay in which SLAMF6-expressing Raji cell lysate and SLAMF6 KO Jurkat T cell lysate were used as immobilized antigens, respectively (Fig 4B).

To assess the biological activity of the bispecific antibody, Jurkat T cells were stimulated with anti-CD3 alone, anti-SLAMF6 alone, anti-CD3 in mixture with anti-SLAMF6, and with the bispecific anti-CD3/SLAMF6 antibody. As we had predicted, increased IL-2 secretion was seen after stimulation with anti-CD3/SLAMF6 as compared with either anti-CD3 alone or anti-CD3 in combination with soluble anti-SLAMF6 (Fig 4C). A dose–response with the use of the bispecific antibody was seen, with greater T cell activation at higher concentrations of anti-CD3/SLAMF6 antibody used for stimulation.

To validate the functional activity of anti-CD3/SLAMF6 in a more physiological system with an intact IS, we used the bispecific antibody to stimulate a co-culture of Jurkat T cells with Raji B cells in the presence of SEE. We again found that the addition of anti-CD3/SLAMF6 increased IL-2 production, almost irrespective of the amount of SEE added (Fig 4D). The increase in IL-2 release seen with the bispecific anti-CD3/SLAMF6 antibody, as compared with anti-CD3 alone, was not seen when anti-SLAMF6 antibody was added as a soluble admixture (Fig 4E). These findings are consistent with the initial observation that it is the clustering of SLAMF6 with CD3, which occurs as a result of the physical bridging by the bispecific antibody, that is required for net T cell activation. In addition, higher concentrations of the anti-CD3/SLAMF6 resulted in greater IL-2 release, suggesting dose–response to the use of this bispecific antibody (Fig 4E).

We therefore show that the novel design of an anti-CD3/SLAMF6–bispecific monoclonal antibody enhances T cell activation greater than either antibody does individually or in combination. Furthermore, a dose-dependent effect exists such that greater

supernatant was harvested and IL-2 levels at different time intervals over 96 h and (E) IFN-y levels at 48 h were analyzed by ELISA. **(F)** Jurkat T cells were stimulated in the presence of brefeldin for 6 h, after which time intracellular IL-2 was analyzed by flow cytometry. This experiment was repeated twice (n = 2). **(G)** Freshly isolated primary CD3 T cells were cultured as above for 120 h. Cell differentiation was analyzed based on cell surface expression of CD45RA and CCR7. A weighed T cell maturation index was calculated as (1*Naïve + 2*Central Memory + 3*Effector Memory + 4*Terminal Effector Memory)/4. This experiment was repeated twice with the average value shown here. **(H)** Cell number was assessed by automated cell counting every 24 h. The experiment was done in triplicate (n = 3). *$P \leq 0.05$ for an unpaired $t$ test.

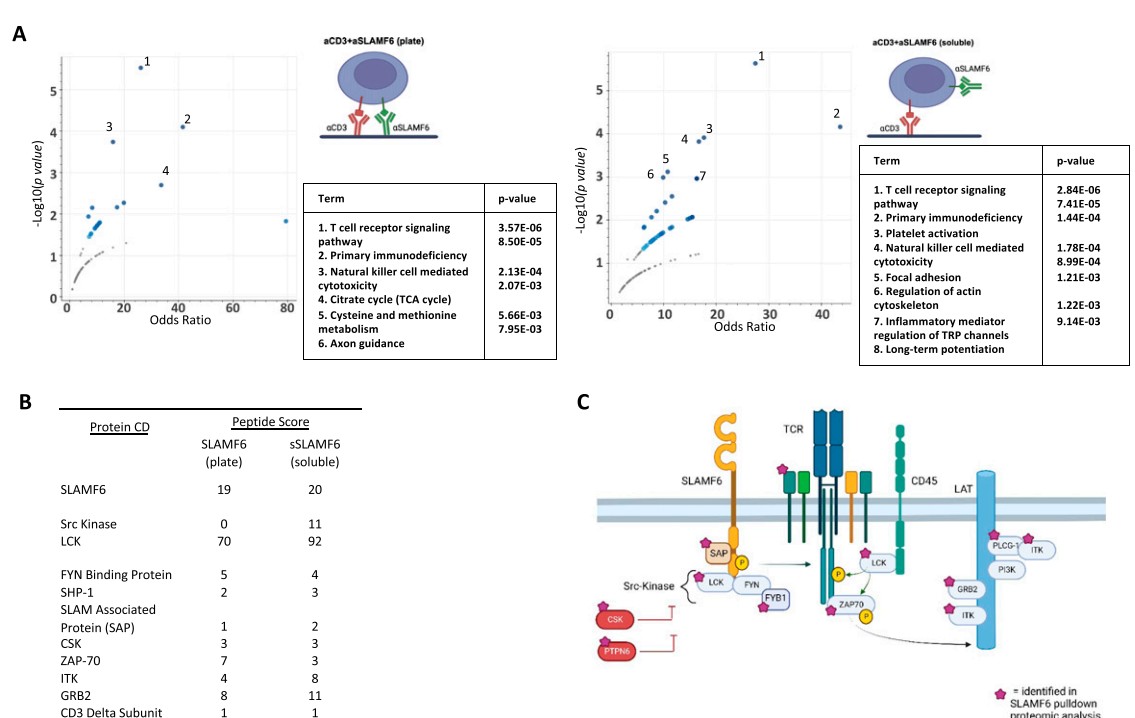

**Figure 2. SLAMF6 and TCR signaling complexes share key mediators.**
V5-SLAMF6-expressing Jurkat T cells were stimulated with plate-coated αCD3 + αSLAMF6 (plate) or plate-coated αCD3 + soluble αSLAMF6 (soluble) for 15 min. The cells were then lysed and the lysate mixed with V5-coupled agarose beads to enrich for V5-tagged SLAMF6 immunoprecipitation. Pull-down lysate proteins were separated by electrophoresis and submitted for mass spectrometry analysis. **(A)** Protein enrichment pathway analysis for the plate versus soluble stimulation conditions was performed. **(B)** Proteins interacting with SLAMF6 were identified, listed by peptide-spectrum match score. **(C)** A schematic demonstrating that the SLAMF6-interacting proteins identified in the immunoprecipitation (marked by a purple star) are known to be essential for proximal TCR signal transduction, emphasizing the interconnection between the two receptors and their signaling interactome. Three independent experimental repeats were performed. (n = 3).

T cell activation occurs at higher concentrations of bispecific anti-CD3/SLAMF6 antibody used.

## T cell stimulation with anti-CD45/SLAMF6–bispecific antibody works predominantly in cis to augment T cell activation

Having shown that a bispecific antibody can augment T cell activation by promoting receptor clustering, we next sought to evaluate whether T cell inhibition can similarly be achieved after forced disruption of receptors' colocalization. Specifically, we hypothesized that SLAMF6-mediated T cell signaling could be inhibited if the SLAMF6 receptor can be tethered away from the IS. We planned to accomplish this using a bispecific anti-CD45/SLAMF6 antibody. CD45 is a receptor protein tyrosine phosphatase expressed on all leukocytes. It is a large glycoprotein of 180–220 kD with a bulky ectodomain region. Although its presence is essential for the initiation of T cell activation, the large CD45 phosphatase is ultimately excluded from the narrow-spaced and matured IS as a result of steric hindrance (7, 8, 9). To spatially segregate SLAMF6 from the CD3, we sought to create a bispecific monoclonal anti-CD45-SLAMF6 antibody that would bind the SLAMF6 receptor, tethering it to CD45 and thereby excluding it from the IS and the CD3 contact zone (Fig 5A).

We designed the anti-CD45/SLAMF6–bispecific antibody as a fusion immunoglobulin of CD45-IgG-hole and SLAMF6-IgG-knob (Table S1 and Fig S3). Binding to the intended targets was validated using an ELISA assay, with the immobilized ectodomain of CD45 peptide and SLAMF6 peptide used as antigen bait (Fig 5B). We next sought to evaluate whether treatment with the anti-CD45/SLAMF6 antibody would interfere with SLAMF6 enrichment in the IS. To visualize this, we used confocal microscopy imaging of GFP-tagged SLAMF6 and OFPSpark-tagged CD45 in a Jurkat–Raji co-culture. Specifically, Jurkat T cells expressing GFP-tagged SLAMF6 were co-cultured with Far Red fluorescently tagged Raji B cells loaded with SEE. Enrichment of SLAMF6 in areas of synapse formation was seen in 79% of the imaged synapses (Fig 5C). Next, we transfected T cells to express GFP-SLAMF6 and OFPSpark-CD45. Once again, we saw SLAMF6 enrichment in the IS, but this time, we could also visualize CD45 exclusion from the synapse (Fig 5C second row). Finally, we pretreated the Jurkat T cells with anti-CD45/SLAMF6 antibody before the co-culture with Raji B cells (Fig 5C third row). Anti-CD45/SLAMF6 treatment resulted in significant decrease in overall number of synapses and in a reduction of SLAMF6 enrichment in the IS. On visual inspection, SLAMF6 localization was similar to that of CD45, with both being absent from the IS. These findings suggest that SLAMF6, restrained by the bulky CD45, was indeed excluded from the IS in mature synapses.

We next sought to evaluate the functional effect of anti-CD45/SLAMF6 on T cell activation. Distinct to our original hypothesis, stimulation of Jurkat T cells with anti-CD45/SLAMF6 antibodies in the presence of anti-CD3 resulted in enhanced T cell activation as compared with either anti-CD3 alone or in combination with

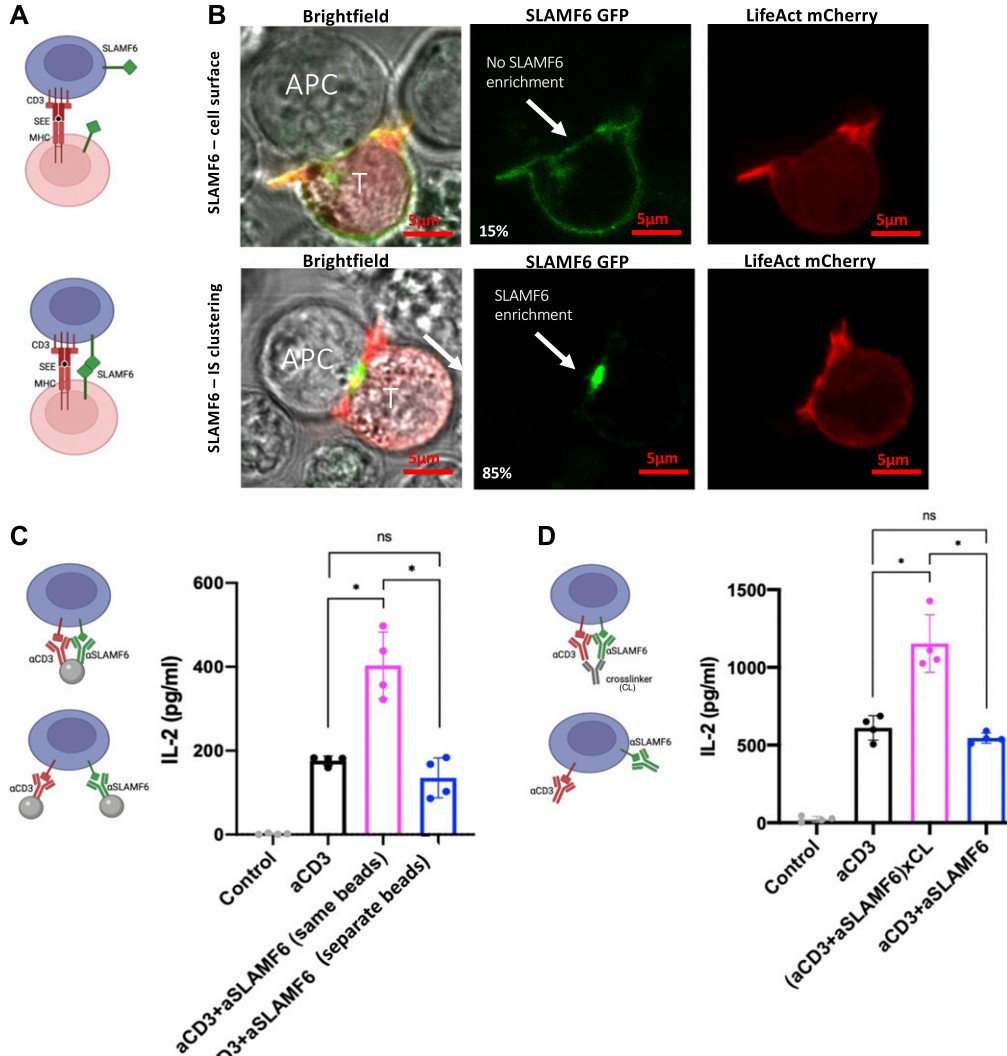

**Figure 3. SLAMF6 clustering in the immunologic synapse (IS) enhances cytokine secretion.**
**(A)** A schematic representation of SLAMF6 in the absence (*top*) or presence (*bottom*) of homophilic receptor ligation in a T-B cell co-culture. **(B)** Jurkat T cells were transfected via nucleofection with LifeAct mCherry and SLAMF6 GFP, then co-cultured with Raji B Cell APCs. Images are representative of at least 30 cells from a least two independent experiments. Scale bar is 5 μm. Actin clearance was defined as a mature IS. **(C)** Jurkat T cells were treated with αCD3-conjugated beads and αSLAMF6-conjugated beads versus αCD3 + αSLAMF6–conjugated beads (both antibodies on the same bead). After 24 h, the supernatant was harvested and IL-2 levels were analyzed by ELISA for at least three independent experiments (n = 3). **(D)** Jurkat T cells were treated with αCD3 + αSLAMF6 or αCD3 + αSLAMF6 + cross-linker for 24 h, after which the supernatant was harvested and IL-2 levels were analyzed by ELISA for at least three independent experiments (n = 3). *$P < 0.05$ for an unpaired $t$ test.

soluble anti-SLAMF6 (Fig 5D). IL-2 release increased with increasing doses of bispecific antibody (Fig 5D). We next repeated the experiment using the Jurkat–Raji co-culture system. Once again, we found that the antibody had a net activating effect on T cells (Fig 5E). At a fixed concentration of 1 μg/ml of anti-CD45/SLAMF6, enhanced T cell activation was seen across increasing doses of SEE added (Fig 5E). A dose–response increase in T cell activation was also seen with increasing levels of the bispecific antibody used, with maximal activation seen at a dose of 10 μg/ml (Fig 5F). On the other hand, stimulation with anti-SLAMF6 (1 μg/ml) and/or anti-CD45 (1 μg/ml) did not significantly enhance the T cell activation as compared with the addition of SEE alone (Fig 5F).

Receptor ligation by the anti-CD45/SLAMF6 has two possible mechanisms of clustering these two receptors: the binding of SLAMF6 and CD45 in trans, between T cells and APC, or in cis, along the T cell surface. In an attempt to better understand how anti-CD45/SLAMF6 resulted in T cell activation despite decreased enrichment of SLAMF6 in the IS, we sought to investigate whether the antibody binds in trans or in cis. Specifically, the aim of this experiment was to compare T cell activation when the anti-CD45/SLAMF6 was added directly into a T and B cell co-culture (trans binding) as compared with adding the anti-CD45/SLAMF6 antibodies to T cells alone (and washing off unbound antibody) before co-culturing with B cells (cis binding). In the first condition, SEE was added to the co-culture to initiate the Jurkat–Raji cross-linking, and this was followed by the addition of the bispecific antibody, thereby preferentially inducing trans binding. Alternatively, in the second condition, we pretreated the Jurkat cells with the bispecific antibody before co-culturing, followed by a wash and subsequent addition of SEE to allow for Jurkat–Raji interaction (Fig 5G). IL-2 levels were increased when anti-CD45/SLAMF6 was added to the Jurkat–Raji co-culture system (trans binding). However, the IL-2 levels were still further increased when the anti-CD45/SLAMF6 antibody was added to Jurkat T cells before the T and B cell co-culture, thus allowing the antibody to bind in cis. This led us to conclude that the anti-CD45/SLAMF6 antibody function in cis along the T cell surface was predominantly responsible for the IL-2 release and may explain the increased T cell activation despite hindered SLAMF6 clustering in the IS.

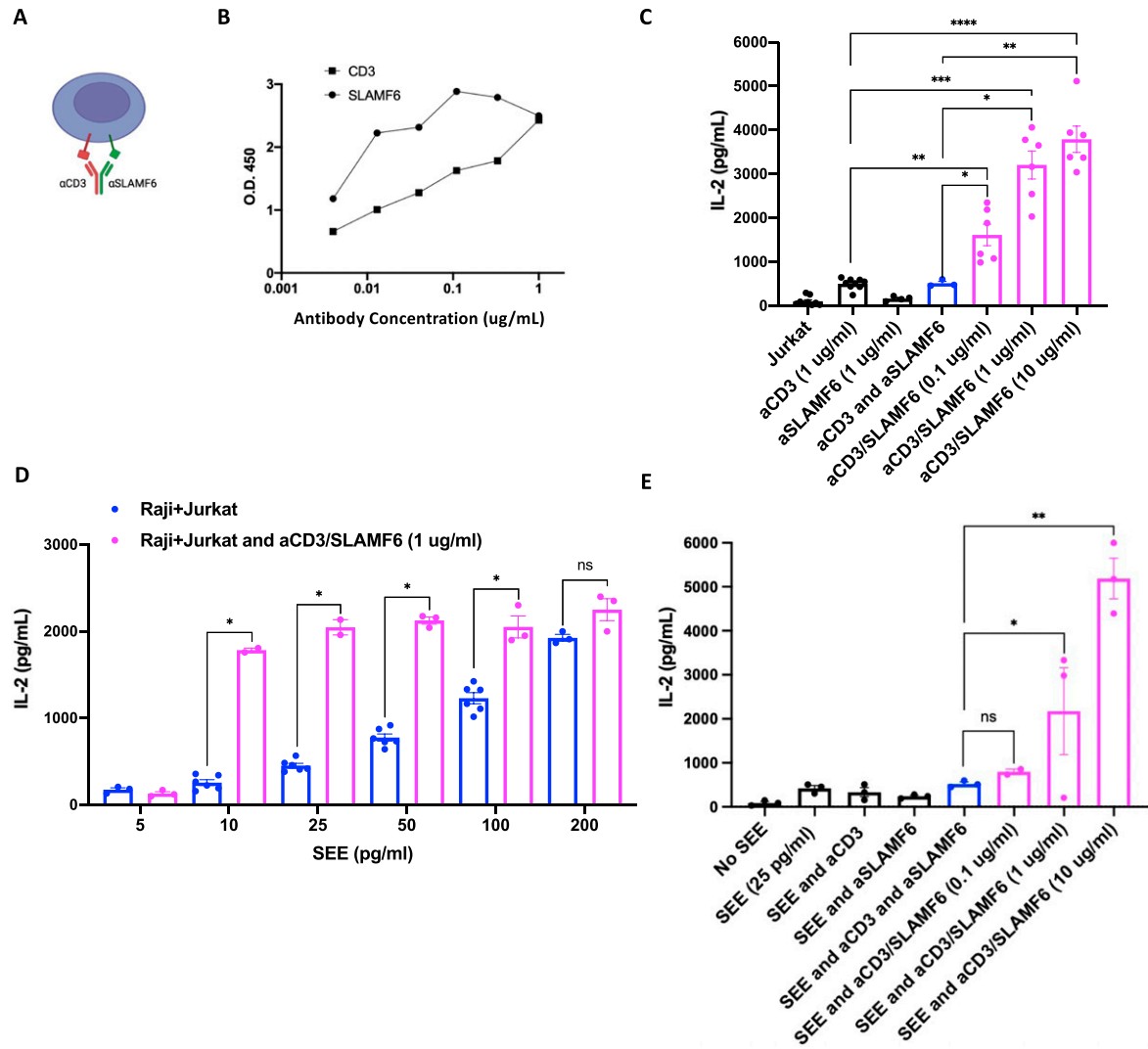

**Figure 4. Anti-CD3/SLAMF6–bispecific antibody clusters SLAMF6 to the CD3 and augments T cell activation.**
**(A)** A schematic representation of anti-CD3/SLAMF6–bispecific antibody (aCD3/SLAMF6) binding and clustering the two receptors together. **(B)** aCD3/SLAMF6 antibody binding was validated using an ELISA assay: αCD3 binding was assessed against immobilized SLAMF6 KO Jurkat T cells, whereas anti-SLAMF6 binding was assessed against immobilized Raji B cells. **(C)** Jurkat T cells were treated with αCD3, αCD3 + soluble αSLAMF6, or aCD3/SLAMF6 at three different concentrations of the bispecific antibody: 0.1, 1, and 10 µg/ml. After 24 h, IL-2 levels were analyzed by ELISA for at least two independent experiments (n = 2). **(D)** Raji B cells were preloaded with increasing concentrations of SEE and co-cultured with Jurkat T cells in the absence (blue) or presence (magenta) of 1 µg/ml of aCD3/SLAMF6 antibody. IL-2 levels were analyzed for at least two independent experiments (n = 2). **(E)** Raji B cells were preloaded with SEE and co-cultured with Jurkat T cells at increasing concentrations of the αCD3/SLAMF6 antibody. IL-2 levels were analyzed for at least two independent experiments (n = 2). *P < 0.05, **P < 0.01, ***P < 0.001 for an unpaired t test.

Our data show that although anti-CD45/SLAMF6 reduces SLAMF6 clustering in the synapses, the bispecific antibody still functions to enhance T cell activation and IL-2 release downstream of the TCR signal. We propose that the antibody exerts its function predominantly via binding in cis along the T cell surface.

### Co-stimulation of primary mononuclear cells with anti-CD45/SLAMF6 augments T cell activation

Observing the stimulatory effect of the anti-CD45/SLAMF6 in an isolated Jurkat T cell system, and a Jurkat–Raji co-culture, we next sought to evaluate the performance of the bispecific antibody in a more complex microenvironment of mixed primary immune cells.

For this, to best mimic the in vivo microenvironment, we chose an ex vivo human PBMC assay. PBMC from healthy donors were activated by the addition of SEE (50 pg/ml). Increasing doses of monovalent anti-CD45/SLAMF6 or bivalent anti-CD45-Ig-SLAMF6 antibodies were added (Fig S4). In the presence of SEE, the addition of the bispecific antibody resulted in increased IL-2 (Fig 6A) and IFN-γ (Fig 6B) release. We observed higher levels of IL-2 release with the bivalent antibody, indicative of its ability to bind more antigen sites, resulting in greater receptor clustering as compared with the monovalent anti-CD45/SLAMF6 antibodies.

SLAMF6 is expressed on a wide variety of immune cells, and we could not be sure if the T cell activation in the PBMC assay was a result of anti-CD45/SLAMF6 binding directly to the T cells or

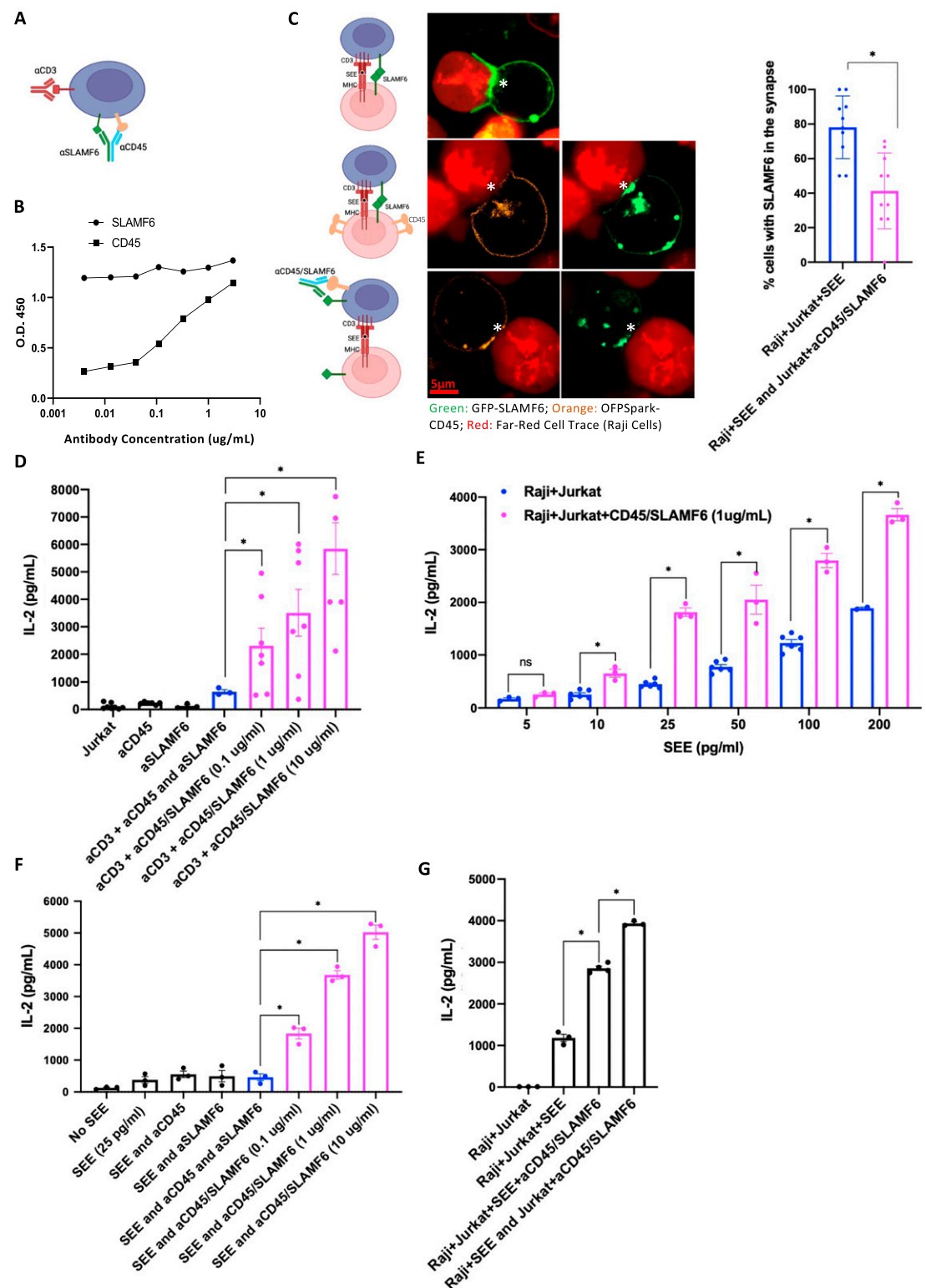

**Figure 5. Anti-CD45/SLAMF6–bispecific antibody inhibits SLAMF6 enrichment in the synapse but still augments T cell activation.**
**(A)** A schematic representation of anti-CD45/SLAMF6 (CD45/SLAMF6) binding to inhibit SLAMF6 clustering with CD3 in the IS. **(B)** αCD45/SLAMF6 antibody binding was quantified using an ELISA assay: αCD45 and αSLAMF6 binding was assessed against immobilized, recombinant ectodomains of CD45 and SLAMF6, respectively. **(C)** Jurkat T cells were transfected with GFP-tagged SLAMF6, and Raji B cells were stained with LifeAct Far Red and preloaded with SEE (2 ng/ml). Jurkat T cells were then co-cultured with Raji B cells for 30 min. Synapse formation with enrichment of SLAMF6 in the IS was visualized (top row). To visualize the distribution of CD45, we next transfected the Jurkat T cells with OFPSpark-tagged CD45 and GFP-tagged SLAMF6. In the Jurkat T–Raji B co-cultures, exclusion of CD45 was coupled with enrichment of SLAMF6 in the IS (middle row). Finally, we pretreated Jurkat T cells with anti-CD45/SLAMF6 10 μg/ml for 15 min (bottom image). Exclusion of CD45 was now associated with a lack of

indirectly as a result of APC activation. This led us to evaluate the effect of anti-CD45/SLAMF6 on primary T cells in culture. After a brief 5-min stimulation of primary T cells with anti-CD3 in the presence or absence of anti-CD45/SLAMF6 antibodies, a trend toward increased phosphorylation of CD3 ζ chain was detected (Fig 6C). After a 24-h incubation under the same stimulation conditions, we observed a significant increase in cell surface CD69 expression and IL-2 and IFN-γ release (Fig 6D).

Thus, we show that anti-CD45/SLAMF6–bispecific antibody is able to enhance T cell activation in an isolated Jurkat–Raji co-culture, in primary T cells, and in a more complex PBMC microenvironment, where it functions to directly activate T cells. This suggests that, in primed T cells, this bispecific antibody has an excitatory effect on T cell activation.

# Discussion

T cell activation is a result of a primary TCR–CD3 complex signal that is fine-tuned by secondary co-stimulatory or co-inhibitory signals transmitted by cell surface receptors that cluster together with the TCR in the contact zones of the IS. SLAMF6 is a transmembrane T cell co-receptor with contradictory reports in the literature as to whether its net effect is to activate or inhibit the TCR signal (3, 4). In this work, we show that the spatial compartmentalization of SLAMF6 with respect to the TCR–CD3 complex in the IS contributes to whether the co-receptor enhances or dampens TCR signal transduction. Specifically, T cell proliferation is inhibited, and there is a trend toward decreased IL-2 release, when the SLAMF6 receptor is engaged but physically removed from the CD3 receptor. On the other hand, when the clustering of SLAMF6 with CD3 is enforced by means of conjugated beads, cross-linking, or bispecific anti-CD3/SLAMF6 antibody, T cell activation is significantly enhanced compared with stimulation with anti-CD3 or anti-CD3 + anti-SLAMF6 combinations. Thus, the spatial distribution of the SLAMF6 receptors in reference to the TCR–CD3 mediates the net signaling effect after receptor ligation. Similar spatial reorganization of co-receptors affecting T cell function has recently been described for another T cell co-receptor, LAG3. Within minutes after LAG3 ligation, the receptor was noted to colocalize with the TCR–CD3 and migrate to the IS, where it exerted its inhibitory function by causing dissociation of LCK from the TCR, dampening TCR activation (10).

The spatial clustering of receptors is essential as it allows for further interaction of the cytoplasmic tails and their associated signaling molecules, propagating the activation signal forward. It is possible that SLAMF6 and/or TCR ligation results in cytoplasmic assembly of signaling units that result in directed movement of the co-receptor along the cell surface. However, our group has previously shown that although vital for T cell activation, the SLAMF6 ectodomain is not necessary to initiate SLAMF6 trafficking to the synapse, suggesting migration signals beyond the ectodomain activation are responsible (6). In the case of SLAMF6, after antigen engagement, the SLAM-associated protein binds to the SLAMF6 intracellular tail and promotes phosphorylation of immunoreceptor tyrosine–based switch motifs by the SRC kinases LCK and FYN. Both SLAM-associated protein and the SRC kinases, in turn, bridge the activation with the TCR–CD3 complex where LCK phosphorylates and activates ZAP70, initiating TCR activation. In our analysis of SLAMF6 interactome, we confirm the association of a number of TCR signaling proteins with the activated SLAMF6 receptor. The clustering of SLAMF6 with the CD3 in the IS allows the intracellular signaling proteins downstream of SLAMF6 to bridge and amplify the TCR signal. On the other hand, we found that when SLAMF6 is removed from the synapse, the intracellular signaling proteins identified in the SLAMF6 interactome are the same as when SLAMF6 is in the IS. We therefore propose that SLAMF6 is able to "steal" signaling proteins, many of which (i.e., LCK) are required in the TCR activation pathway. In such instances of SLAMF6 being removed from the CD3, the net effect is inhibition after SLAMF6 ligation. The concept of inhibiting TCR signaling by removing LCK from the tail of the CD3 complex has been reported recently for the inhibitory receptor LAG3 (10). We conclude that the clustering or removing of SLAMF6 from the TCR–CD3 complex modulates cell activation by either contributing or removing key signaling proteins essential for TCR activation.

Bispecific antibodies are designed to simultaneously recognize two different antigen-binding sites. Trans binding can bridge cells together, whereas cis binding can bring the antigen-expressing moieties closer together in physical space along the cell surface membrane. In cancer immunology, some bispecific antibodies bind in trans to simultaneously engage tumor-associated antigens and CD3, augmenting the anti-tumor response by physically linking tumor cells to T cells (11, 12). Alternatively, other bispecific antibodies bind in cis; an example of which is the targeting of PD-1 with the CD45 phosphatase, resulting in reduced PD-1 phosphorylation and enhanced T cell activity (13). In this work, we bioengineered two bispecific antibodies, anti-CD3/SLAMF6 and anti-CD45/SLAMF6, targeting the CD3 and the SLAMF6 receptors and CD45 and SLAMF6 receptors, respectively. We hypothesized that anti-CD3/SLAMF6, by bridging the two receptors together, would enhance T cell activation. Indeed, we found that stimulation of T cells with

enrichment of SLAMF6 in the IS (bottom row). Images are representative of at least 40 cell conjugates per each experimental condition from two independent experiments. Scale bar is 5 μm. Percent of cell conjugates with SLAMF6 enrichment in the IS was quantified; results are summarized in the bar graph. **(D)** Jurkat T cells were treated with αCD3 and αCD45/SLAMF6 at three different concentrations: 0.1, 1, and 10 μg/ml of the bispecific antibody. After 24 h, IL-2 levels were analyzed by ELISA. **(E)** Raji B cells were preloaded with different concentrations of SEE and co-cultured with Jurkat T cells in the absence (blue) or presence (magenta) of 1 μg/ml of αCD45/SLAMF6 antibody. IL-2 levels were analyzed for at least three independent experiments (n = 3). **(F)** Raji B cells were preloaded with SEE and co-cultured with T cells at increasing concentrations of αCD45/SLAMF6 antibody. IL-2 levels were analyzed for at least three independent experiments (n = 3). **(G)** Raji B cells and Jurkat T cells were co-cultured either in the presence of, or after T cell pretreatment with, αCD45/SLAMF6. Specifically, in the first experimental condition, αCD45/SLAMF6 was added as Jurkat T–Raji B conjugates formed (supporting in trans antibody ligation), whereas in the second experimental condition, Jurkat T cells were pretreated with αCD4/SLAMF6 for 30 min, washed, and subsequently co-cultured with the Raji B cells, supporting in cis antibody ligation on T cells before addition of the B cells. IL-2 levels were analyzed for at least three independent experiments (n = 3). *P < 0.05 for an unpaired t test.

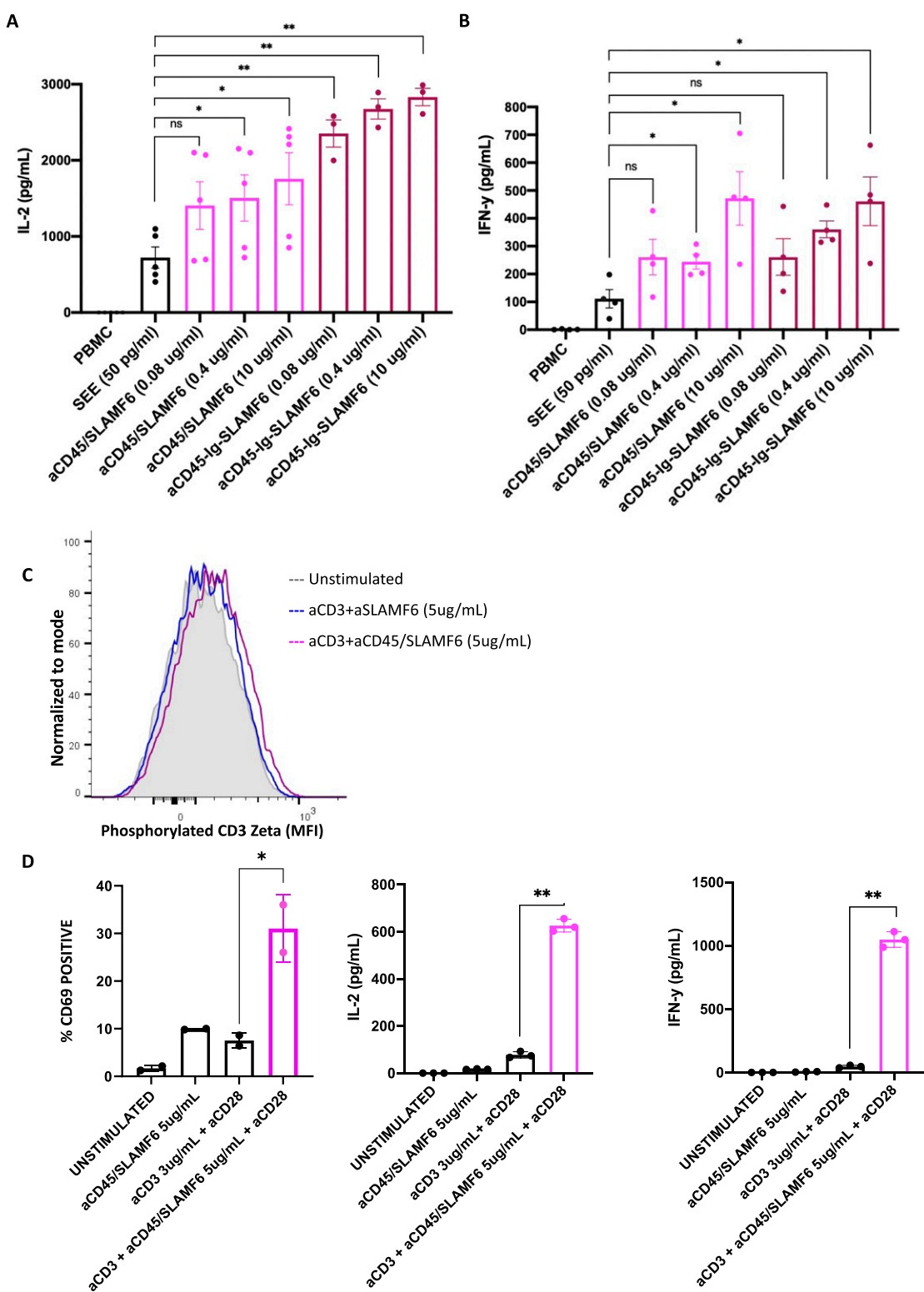

**Figure 6. Anti-CD45-SLAMF6 antibody enhances T cell response in primary human cell–based assay.**
PBMCs were treated with SEE and three different concentrations of either the monovalent (αCD45/SLAMF6) or the bivalent (αCD45-Ig-SLAMF6) anti-CD45/SLAMF6–bispecific antibody. **(A, B)** After 24 h, (A) IL-2 and (B) IFN-γ levels were measured by ELISA. The results of at least three independent experiments (n = 3) are shown. **(C, D)** Primary human CD3-positive T cells were isolated from whole blood and cultured with anti-CD3 in the presence αCD45/SLAMF6. **(C)** After 5 min, the cells were analyzed for phosphorylation of CD3 ζ chain using flow cytometry. **(D)** After 24 h, the cells were analyzed for CD69 expression, and the supernatant was analyzed for IL-2 and IFN-γ release. *P < 0.05, **P < 0.01 for an unpaired t test.

anti-CD3/SLAMF6 augments activation to a greater extent than either anti-CD3 or anti-SLAMF6 alone or in an admixed combination. Thus, we suggest that antibodies targeting in cis clustering of T cell co-receptors such as SLAMF6 may have immunotherapeutic potential to regulate T cell function in disease. Furthermore, although bispecific antibodies that activate CD3, such as anti-CD3/SLAMF6, are fraught with the risk of nonspecific T cell activation and adverse inflammatory immune-related effects, bispecific antibodies targeting co-receptor function, such as anti-CD45/SLAMF6, would be expected to minimize this risk. Specifically, unlike anti-CD3/SLAMF6, anti-CD45/SLAMF6 would not be expected to activate resting T cells in the absence of TCR engagement and may therefore be a safer therapeutic intervention to target only the activated T cells, minimizing off-target effects.

Similarly, and somewhat unexpectedly, we found that anti-CD45-SLAMF6 antibody can also enhance T cell activation. CD45 is a transmembrane protein tyrosine phosphatase expressed on most hematopoietic cells. CD45 localization on T cells, although critical for cell activation, is such that it is excluded from the IS because of its bulky steric hindrance (8, 9). In addition, as mentioned above, bispecific antibodies in development rely on the CD45 phosphatase to colocalize and dephosphorylate the associated co-receptors through in cis interaction (13). We thus hypothesized that by tethering the SLAMF6 to the large and bulky ectodomain of CD45 phosphatase, we could exclude SLAMF6 from the IS because of steric hindrance, resulting in net inhibition of T cell function (8, 9). Contrary to our expectation, stimulation with anti-CD45/SLAMF6 enhanced rather than inhibited T cell activation. We repeatedly found that stimulation with anti-CD45/SLAMF6 resulted in net activation effect on T cell function. Several explanations exist to explain our findings. *First*, CD45 is critical for T cell survival, and CD45-deficient mice have a block in T cell development that is attributed to the ability of CD45 to dephosphorylate a negative regulatory tyrosine on LCK. Thus, the CD45 functions to activate LCK kinase, priming it for recruitment to the SLAMF6 and TCR, further propagating the activation signal (14, 15, 16). Therefore, the proximity of CD45 to SLAMF6 that was achieved with our bispecific antibody may have resulted in augmented LCK activation, promoting SLAMF6 phosphorylation and overall T cell activation. *Second*, microscopy imaging shows that although SLAMF6 clustering at the IS is reduced after treatment with anti-CD45/SLAMF6 in the mature synapse, it was not completely abolished. Indeed, initial recruitment of CD45 to the IS is required for cell signal propagation, with expulsion of CD45 because of steric hindrance occurring only subsequently to its initial recruitment (14). *Finally*, it is possible that although we intended our bispecific antibody to bind in cis, some degree of in trans binding occurred, promoting cell clustering and net activation.

Taken together, we show that SLAMF6 activity is dependent on its localization along the cell surface with respect to the TCR site. Activation of SLAMF6 in proximity to the CD3 shows that SLAMF6 is an activating T cell co-receptor. SLAMF6 interacts with a number of proximal TCR signaling proteins, suggesting translocation to the TCR site after receptor ligation. Enhanced T cell activity can be achieved by stimulating with an anti-CD3/SLAMF6–bispecific antibody, with a dose-specific response seen in our assays. On the other hand, when SLAMF6 is removed from the TCR site, it is able to "steal" with it

many of the essential proximal signaling proteins that are required for TCR signal propagation. Thus, a neutralizing, and sometimes inhibiting, effect can be seen. We conclude that spatial localization of T cell co-receptors along the T cell surface may be a novel mechanism to target T cell activity in disease.

Our work has several limitations. We show that despite differences in cell activation, mass spectrometry analysis of the SLAMF6 interactome when clustered with CD3 as compared with when it is removed from CD3 is not remarkably different. We explain this by a "steal" phenomenon where SLAMF6 localization away from the CD3 removes essential signaling proteins away from the CD3, dampening T cell signaling. A limitation to our explanation is that we were not able to detect CD3 in our IP; thus, we could not prove differential CD3 association between the two conditions. Another limitation of the IP is that we did not have results for protein phosphorylation states. Yet another limitation is that although we show that the synthesized bispecific antibodies in this work are functional with the promise to enhance T cell activity, we were not able to achieve T cell inhibition by removing SLAMF6 from the IS. Although we believe that this was a result of in cis interaction between CD45 and SLAMF6, the exact mechanism remains to be evaluated in future works. Finally, we show functional activity of the synthesized antibodies in biochemical, co-culture, and PBMC experiments, but we were not able to evaluate the antibody function in animal models. Future work will focus on obtaining humanized mice and/or synthesizing mouse-specific antibodies to better evaluate the therapeutic potential of bispecific SLAMF6 activating antibodies in tumor immunology.

# Materials and Methods

### General reagents

RPMI medium 1640, DMEM, PBS, and FBS were purchased from Life Technologies. SEE was acquired from Toxin Technology. BCA protein assay kit was purchased from Thermo Fisher Scientific (# 23227).

### PBMC isolation

10–15 ml of whole blood was collected into EDTA tubes from healthy volunteers who provided informed consent (IRB-AAAB3287). Mononuclear cells were isolated using Lymphoprep density gradient centrifugation (STEMCELL Technologies).

### Cells

Jurkat T cells and Raji B cells were obtained from the American Type Culture Collection. The cells were cultured in RPMI 1640 medium supplemented with 10% FBS and 100 U/ml penicillin and streptomycin. Primary CD3[+] T cells were isolated from PBMCs using the EasySep Human T Cell Isolation Kit (# 17951) and were grown in RPMI 1640 medium supplemented with 10% FBS and 100 U/ml penicillin and streptomycin, nonessential amino acids (2 nM), and L-glutamine (2 mM). HEK293T cells were obtained from American Type

Culture Collection and cultured in DMEM media supplemented with 10% FBS and 100 U/ml penicillin and streptomycin.

## SLAMF6 knockout Jurkat T cells

SLAMF6 was knocked out (KO) in Jurkat T cells by CRISPR-Cas9 using two of the lentiCRISPR v2 plasmids purchased from GenScript. Two sets of lentiviral particles were generated as before, with each set containing one of the different lentiCRISPR v2 plasmids. Viral particles were transduced by centrifugation, and cells were selected with puromycin.

## OFPSpark-CD45- and GFP-SLAMF6-expressing Jurkat T cells

SLAMF6-GFP fusion expression constructs were generated through PCR amplification and cloning of SLAMF6 (DNASU #HsCD00446754) into pEGFP-N1 vector (Invitrogen). OFPSpark-CD45 DNA constructs were obtained from Sino Biologics (#HG10086-ACR). DNA expression constructs were introduced into the SLAMF6 KO Jurkat T cells by nucleofection according to the optimization protocol (Lonza Nucleofector II).

## V5-SLAMF6 stable Jurkat T cell line

V5-SLAMF6 expression construct was purchased from DNASU (#HsCD00938682). Plasmid DNA was isolated using the maxi plasmid purification kit (QIAGEN). For lentiviral production, plasmid DNA expressing V5-tagged SLAMF6 was co-transfected with pMD2G envelope and psPAX2 packaging plasmids in HEK293T cells using SuperFect transfection reagent (QIAGEN). 2 million Jurkat T cells were lentivirus transduced by spinoculation at 800*g* for 30 min at 32°C. Blasticidin selection was used for the generation of V5-SLAMF6-expressing Jurkat clones. SLAMF6 expression was confirmed by flow cytometry analysis using PE-fluorescent anti-SLAMF6 antibody staining (#317207; BioLegend).

## Antibodies and stimulation

Anti-CD3 (UCHT1, #300465; BioLegend) and anti-SLAMF6 (#317202; BioLegend) antibodies were used for stimulation. For immobilized stimulation, plates were coated with anti-CD3 (3 µg/ml, unless otherwise indicated) and anti-SLAMF6 (5 µg/ml, unless otherwise indicated); for soluble stimulation, the antibodies were added in suspension at the same concentration. Antibody-coupled Dynabeads (#14311D; Invitrogen) were created according to manufacturer's protocol; anti-CD3/IgG isotype, anti-SLAMF6/IgG Iso, and anti-CD3/anti-SLAMF6–conjugated beads were used for stimulation in a final suspension of 0.1–0.2 mg beads per ml.

Bispecific antibodies used in this study were made by a rational structure-guided approach that resulted in a set of substitutions that were reported to lead to over 90% heterodimers with a high thermal stability. The single-chain fragment variable of the anti-SLAMF6 Ab and the anti-CD45 Ab was linked to the fragment crystallizable region (Fc) of the two heavy chains including T350V, T366L, K392L, and T394W mutations in the first Fc chain and T350V, L351Y, F405A, and Y407V mutations in the second Fc chain. The bispecific antibodies were generated by subcloning the binding domains of OKT3, SLAMF6, and CD45 into pVax1 vector (#V26020; Thermo Fisher Scientific) using HindIII and XbaI restriction sites (sequences are included in Table S1). Expi293 cells (#A1435101; Thermo Fisher Scientific) were grown with serum-free expression medium until confluency on 37°C $CO_2$ shaker (Benchmark Scientific ORBi-SHAKER #BT4001) at 110 rpm. 200 million cells were transfected with Gibco ExpiFectamime Transfection Kit (#A14524; Thermo Fisher Scientific) following manufacturer's recommendations. On day 5 post-transfection, supernatant was collected and antibodies were separated using protein A agarose beads (#20333; Thermo Fisher Scientific) and elution buffer (#21004; Thermo Fisher Scientific). Antibodies were supplemented with 1 M HEPS buffer, and concentration was determined by OD measurement at 280 nm and by running PAGE gel against 1 µg of BSA as a control.

## Cell proliferation assay

Jurkat T cells were activated with soluble anti-CD3 or anti-CD3 + anti-SLAMF6 and cultured for 72 h. The number of cells was assessed by automated counting (Invitrogen Countess II) in the presence of trypan blue. Primary CD3+ T cells were isolated, stained with 1 µM of carboxyfluorescein succinimidyl ester (CFSE; #79898; BioLegend) then activated with immobilized anti-CD3 or anti-CD3/anti-SLAMF6 for 120 h. Cells were then assayed for proliferation via flow cytometry after a period of 5 d under stimulatory conditions.

## Flow cytometry analysis

At the completion of stimulation, cells were collected and surface stained for CD4-FITC (#300506; BL) or CD4-AF700 (#300526; BL), CD8-BV605 (#301040; BL), CD69-BV421 (#310930; BL), CD25-FITC (#302604; BL), PD1-BV711 (#329928; BL), CD45-BV421 (#304129; BL), and CCR7-APC (#353213; BL). Intracellular staining for IL-2-PE (#500307; BL) was performed after fixation (#420801; BL) and permeabilization (#421002; BL). Intracellular staining for phosphorylated CD3 ζ-AF647 (#558489; BD) was performed after fixation and above and permeabilization (#76344; BL). Flow cytometry was performed on the BD LSRII and analyzed with FlowJo v10.8.1 software.

## Confocal microscopy

To evaluate conjugate formation by confocal microscopy, Jurkat T cells were transfected with GFP-SLAMF6 and/or mCherry-LifeAct-7 (#54491; AddGene)–expressing constructs. Raji B cells were pre-stained with CellTrace Far Red dye (#C34572, 1:1,000 dilution for 20 min in PBS; Invitrogen). In some conditions, $1 \times 10^6$ Jurkat T cells were pretreated with either anti-SLAMF6 or anti-CD45/SLAMF6 as per the experimental design. $1 \times 10^6$ Raji B cells were coated with 2 mg/ml SEE in FCS-free RPMI for 2 h. Next, $2–3 \times 10^5$ in 100 µl Jurkat T cells were mixed with $2–3 \times 10^5$ in 100 µl Raji B cells. The co-culture was placed on a glass bottom culture dish (MatTek Corporation) and rested for 15 min to allow conjugates to form. Subsequently, confocal images from each stimulation were acquired on Zeiss LSM 900 confocal microscope and analyzed with ZEN Blue software.

## The ELISA

To determine the concentration of secreted proteins after stimulation, human IFN-γ (#430101; BioLegend) and human IL-2 (#431801; BioLegend) detection kits were used. To quantify anti-CD45 and anti-SLAMF6 antibody binding, polystyrene high binding microplates (Corning) were coated with immobilized CD45 (#14197-H08H; SinoBiological) or SLAMF6 (#11945-H08H; SinoBiological) recombinant ectodomain proteins, respectively. To quantify anti-CD3 and/ or anti-SLAMF6 antibody binding to CD3, immobilized SLAMF6 KO Jurkat T cell lysates were used as antigen bait. Primary antibody binding was detected using a secondary HRP goat antibody recognizing human Fc (#SSA001-200; Sino-Biologic).

## Immunoprecipitation

Jurkat T cells expressing V5-tagged SLAMF6 were stimulated with immobilized anti-CD3 and anti-SLAMF6 as described above. $30 \times 10^6$ Jurkat T cells from each condition were lysed in an IP lysis buffer (1% NP-40; 25 mM Tris, pH 8.0; 150 mM NaCl; 1 mM EDTA; 5% glycerol) containing protease and phosphatase inhibitors. Cell lysates were incubated with anti-V5 monoclonal antibody coupled to agarose beads to enrich V5-tagged SLAMF6, according to the manufacturer's protocols (#PM003-8; MBL). Pull-down lysates were separated by Tris-glycine gels and submitted for mass spectrometry analysis at Quantitative Proteomics and Metabolomics Center at Columbia University and NYU Langone's Proteomics Laboratory. Proteins identified were analyzed in the context of known signaling pathways, identified using public databases (uniprot.org and reactome.org). Protein–protein interaction was explored using STRING analysis (https://string-db.org/).

## Statistics

Values are reported as mean ± SD or median ± 95% CI. Statistical analyses were performed using paired *t* test for normally distributed data and Mann–Whitney U test for skewed data. All statistical analyses were performed using GraphPad Prism (version 8.0). Illustrations were created using BioRender.

## Ethics statement

This work did not involve human patients or animals.

# Supplementary Information

# Acknowledgements

This work was supported by grants from the NIH (AI125640, CA231277, AI150597) and the Cancer Research Institute and the Lisa M Baker Autoimmunity Innovation Fund. We would like to thank Dr. Joan Bathon for her support of this project.

## Author Contributions

Y Gartshteyn: conceptualization, data curation, formal analysis, validation, investigation, visualization, methodology, and writing—original draft, review, and editing.
AD Askanase: conceptualization, data curation, supervision, investigation, and writing—review and editing.
R Song: investigation, methodology, and writing—review and editing.
S Bukhari: data curation, formal analysis, investigation, methodology, and writing—review and editing.
M Dragovich: investigation, methodology, and writing—review and editing.
K Adam: investigation, methodology, and writing—review and editing.
A Mor: conceptualization, resources, data curation, formal analysis, supervision, funding acquisition, validation, investigation, visualization, methodology, and writing—review and editing.

## Conflict of Interest Statement

The authors declare that they have no conflict of interest.

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
