## [Reviewer comments · Life Science Alliance]

Life Science Alliance

SLAMF6 compartmentalization enhances T cell functions.

Yevgeniya Gartshteyn, Anca Askanase, Ruijiang Song, Shoiab Bukhari, Matthew Dragovich, Kieran Adam, and Adam Mor

DOI: <https://doi.org/10.26508/lsa.202201533>

Corresponding author(s): Yevgeniya Gartshteyn, Columbia University Medical Center

Review Timeline:

Submission Date:	2022-05-25
Editorial Decision:	2022-06-27
Revision Received:	2022-10-25
Editorial Decision:	2022-11-22
Revision Received:	2022-11-28
Accepted:	2022-11-29

Scientific Editor: Novella Guidi

Transaction Report:

June 27, 2022

Re: Life Science Alliance manuscript #LSA-2022-01533

Dr. Yevgeniya Gartshteyn
Columbia University Medical Center
Medicine
630 West 168th Street
New York, NY 10032

Dear Dr. Gartshteyn,

Thank you for submitting your manuscript entitled "SLAMF6 compartmentalization enhances T cell functions." to Life Science Alliance. The manuscript was assessed by expert reviewers, whose comments are appended to this letter. We invite you to submit a revised manuscript addressing the Reviewer comments.

Thank you for this interesting contribution to Life Science Alliance. We are looking forward to receiving your revised manuscript.

Sincerely,

B. MANUSCRIPT ORGANIZATION AND FORMATTING:

Reviewer #1 (Comments to the Authors (Required)):

This paper is focused on mechanisms involved in the regulatory effects of self-binding immune receptor SLAMF6 in T lymphocytes, and offers a spatial explanation for the contradicting effects attributed to this receptor in past reports. Basically, the paper claims that bispecific anti-CD3/SLAMF6 antibodies promote SLAMF6 clustering with CD3 and thus enhanced T cell activation.

To show this several experiments repeat a basic structure of T cells/Jurkat cells functional assessment mainly via cytokine secretion. To enhance clustering of CD3 and SLAMF6, the authors use plate-bound aCD3 and aSLAMF6, or beads coated by the antibodies. In addition to cytokine secretion, CFSE is used in Jurkat cells for proliferation, and mass spectrometry to read pull-down protein capture. Then, the bispecific anti-CD3/SLAMF6 antibodies is used to enhance CD3-SF6 proximity.

Here are some comments on data sections:

SLAMF6 mediated T cell activation requires spatial co-localization of SLAMF6 and CD3 along the cell membrane- Data presented in Figure 1 show primary CD3 T cell activation and compare plate-bound CD3 with SLAMF6-targeting Abs to soluble anti-SLAMF6 Ab. The first enhance cytokine secretion, while the latter reduces it. Data would have been more convincing if the T cells would secrete higher IL-2 or IFN γ under CD3 stimulation (given with 3 μ g/ml, which is high enough), but nevertheless, the data in favor of SF6/CD3 synergy is consistent, with a p value of 0.05, to support the working hypothesis.

SLAMF6 co-immunoprecipitation (co-IP) assay identifies downstream proteins that bridge SLAMF6 and TCR signaling. Network derived of a pull-down experiment that generally show higher peptide scores of SF6-related adaptors and binders with the cluster enhancing stimulation. The immune precipitation shows a non-differential pulldown of essential T cell signaling proteins in both the soluble and plate SLAMF6 conditions. The authors explain this similarity as the reason why soluble SF6 binding is actually weakening the CD3-related stimulation. Any other reference to signaling, including phosphor-signals is not provided. The reference to pull-down as reflecting short-term signals is insufficient, considering the strong background of this assay- and the authors report of only a series of signal proteins while ignoring others.

SLAMF6 mediated T cell activation is enhanced when SLAMF6 and CD3 cluster in the immunologic synapse. This section adds that the need for proximity between SF6 and CD3 is relevant to the immune synapse. The interacting cells are Jurkat- as effectors, and Raji- as APCs. A single synapse's focal microscopy is shown and cytokine release from the Jurkat cells confirms data from figure 1. Jurkat cells vary in their sensitivity to anti-CD3 surface domain binding and for activation, the use of PMA and ionomycin is required.

T cell stimulation with anti-CD45/SLAMF6 bispecific antibody works predominantly in cis to augment T cell activation This construct was design to be a negative control for the previous bispecific Ab, hypothesizing that it would spatially segregate SLAMF6 from the CD3. Given that this hypothesis fell- no reduced reactivity occurred, the authors suggest that it mainly act in cis, and used the reagent in a mixed population of cells.

Strengths:

- 1- The experiments described yield consistent data, taking into consideration that they scan only a limited repertoire of T cell functions.
- 2- The bispecific anti-CD3/SLAMF6 antibodies are a new idea and validated in cytokine release assays
- 3- The simple yet elegant methodologies that were used to differentiate CD3-SF6 clustering-enhancement versus a non-clustering situation of a soluble antibody.

Weaknesses:

- 1- The experimental design focuses mainly on cytokine secretion with very few- if at all, additional surveys of other aspects of T cell activation.
- 2- Phenotypical markers of activation, cytotoxicity markers, level of maturation are all not included.
- 3- Deconstruction of primary CD3 T cells into CD4 and CD8 is not provided.
- 4- The level of expression of SLAMF6 on the APC Raji cells- is not shown, which is a critical element in determining the role of SF6 when there is trans-activation.

- 5- A single micrograph is shown to support the idea that SF6 clusters in the IS as a general phenomenon
 - 6- Some of the data relies on Jurkat cells, in which the CD3/TCR complex is disrupted
 - 7- No data is shown what happens- as in real life- when the TCR is triggered via epitope-MHC conjugates and not by a non-specific anti-CD3-induced receptor cross-binding
 - 8- The data on the bifunctional SF6-CD45, which was originally meant to remote SF6 from CD3 and be used as a negative control, is not contributing to the main theme.
 - 9- The basic level of almost 0 activation by SEE indicates that there is a co-inhibitory interaction between the Raji and the T cells, otherwise I would expect a basic response of the CD4 component of the T cell to the SEE.
- To summarize: while the basic hypothesis is interesting and the bi-specific construct data is new, the general picture is based on restricted survey of T cell functions and leave significant vacancies in critical aspects of immune synapse-induced T cell activation

Reviewer #2 (Comments to the Authors (Required)):

Gartshteyn et al. describes their research on the role of SLAMF6 in T cell activation and the regulation of SLAMF6 activity by bispecific antibodies. This is an interesting work that potentially paves way to a novel immunoregulatory drug target molecule. However, there are some major and minor concerns that need to be addressed before the manuscript can be considered for publication.

Major points

1. The authors used a soluble anti-SLAMF6 antibody for the sequestration of SLAMF6 from IS. Please provide a detailed explanation about how the soluble antibody forces SLAMF6 out of IS, as opposed to uniform distribution (i.e. not concentrated in IS) over the entire cell surface, in which case the result would be similar to an isotype control antibody.
2. A detailed analysis of Fig. 2A is needed. It is not clear what "a non-differential pulldown of essential T cell signaling proteins" means, when there are more than twofold difference between the "plate" and the "soluble" sets for some of the proteins (Src, SAP, ZAP70, etc.). In my opinion this is the most important molecular-level data supporting the author's claim that SLAMF6 is stimulatory when colocalized in IS but inhibitory when kept away from it.
3. Please provide some comments (maybe in discussion section) about what causes SLAMF6 colocalize in IS during T cell activation.
4. In page 14, the authors stated that "anti-CD45/SLAMF6 would not be expected to activate resting T-cells in the absence of TCR engagement". Please provide data supporting this claim (i.e. T cells treated by anti-CD45/SLAMF6 alone vs. aCD3 + anti-CD45/SLAMF6).
5. Bispecific antibodies (bsAbs) used in this study were made by knobs-into-holes (KiH) technique. Please provide details of bsAb generation in the methods section. Specifically, how did authors solve the problem of the random pairing between heavy and light chains? In antibody engineering perspective, this usually is solved by common light chain approach, or by more elaborate techniques such as CrossMab. Alternatively, a mixture of randomly paired KiH antibodies may be used for the experiments. In any case, it needs to be described in detail how the bsAbs were constructed and produced.
6. Could the authors elaborate more on the "two predominant patterns of SLAMF6 expression" (page 7 and Fig. 3B)? Data are provided but explanation is lacking.

Minor points:

1. It is not clear at all what "bivalent anti-CD45-Ig-SLAMF6 antibody" is. Maybe a cartoon showing the structures of bsAbs used in this study would be helpful.
2. bsAb notations are not consistent. Usually it is "anti-CD3/SLAMF6", but in some instances "anti-CD3-SLAMF6" (slash vs. hyphen). Probably the most formal and standard way is using a cross sign (i.e. a CD3 x SLAMF6 bispecific antibody, without "anti-").
3. Full terms for abbreviations should be provided at the first appearance. For example, SEE (staphylococcus enterotoxin E) is used throughout the manuscript, but the full term is not explained until page 16 (materials and methods).
4. V5/GFP tagged SLAMF6 expressing Jurkat T cells -> Jurkat T cells expressing V5/GFP-tagged SLAMF6 (page 5 and 9).
5. Something is missing in the sentence "Indeed, initial recruitment of both CD45 and in early IS formation is required for cell signal propagation" (page 14).
6. Page 15: the bsAbs in this study are not "humanized". A humanized antibody refers to an antibody originating from animal (e.g. mouse) and later engineered so that its amino acid sequence closely resembles that of human antibodies, in an attempt to minimize immunogenicity when administered to human subjects as a therapeutic agent. Therefore "humanized" has nothing to do with "binding to human proteins".

Reviewer #3 (Comments to the Authors (Required)):

Brief description of paper:

The paper describes the effects of SFLAMF6 localisation on CD3 induced TCR activation on T cell / Jurkat T cell or APC-T cell

activation. The experimental data shows compelling evidence that SLAMF6 localisation within the immune synapse provides strong synergistic activation. While this phenomenon has been previously published by the authors, the creation and validation of the bi-specific CD3/SLAMF6 antibody does present a substantial application advancement. The unexpected activation response obtained with the application of the second bi-specific CD45/SLAMF6 antibody, while intriguing, would however require further confirmation on its targeting site (outside or within the immune synapse, and the cellular target). On a lesser note, the impact of SLAMF6 localisation outside the immune synapse while novel, is not entirely clear across the experiments. In some instances, SLAMF6 localisation outside the immune synapse represent an inhibitory response (T cell proliferation), in others, it is a loss of synergistic activation (IL-2). This could be due to the difference in sampling time points (24, 48 versus 120 hours), and would require further experiments or clarifications.

Main points of paper:

Section: SLAMF6 mediated T cell activation requires spatial co-localization of SLAMF6 and CD3 along the cell membrane.

The data is supportive of that co-localisation synergistic TCR activation with SLAMF6 using the plate bound anti-CD3/SLAMF6. On a lesser note, the ELISA response from plate bound anti-CD3 with soluble anti-SLAMF6 needs more experimental validation or a re-evaluation on their statement. The expected time frame for experiment would be 1-3 months, depending on reagent availability.

1. In Figure 1Bii, the authors compared several conditions experimenting with plate bound/soluble anti-SLAMF6 with plate bound anti-CD3 on T cell proliferation. The presence of soluble anti-SLAMF6 appears to reduce T cell proliferation by plate bound anti-CD3. Did the authors checked if the reduction of T cell proliferation could be caused by unspecific FC-receptor binding of the soluble anti-SLAMF6 ? This can be determined by adding the respective specific antibody FC iso-type control for soluble anti-SLAMF6, or alternatively using soluble Fab anti-SLAMF6.

2. In Figure 1C, there was a trend on the inhibitory effect of segregating SLAMF6 on the secretion of IL-2, but it was not significant. Notably, the T cell proliferation assay proceeded for 120hrs, while the ELISA supernatant was harvested within 48hrs. Would an analysis of the cytokines at the 120hrs time point be a better reflection of the observation seen in T cell proliferation ? However in Figure 1D, the impact of using soluble SLAMF6 with plate anti-CD3 had a synergistic trend compared to anti-CD3 alone for IFN γ secretion. Why is the trend reverse for IFN γ secretion for this condition ? Additionally for Figure 1C-1D, the figure legends mentioned that technical triplicates were done for each condition, yet it can be clearly seen in the diagram that certain conditions had more than 3 data points. I am not sure if this is a graphical issue or otherwise. If the data points are not equivalent across the conditions, this could have affected the statistics, perhaps the p-value could be properly re-calculated.

3. In the summary for Figure 1, the authors need to differentiate between the interpretations that the difference is (a) an inhibitory effect or (b) loss (dampening) of synergistic effect when segregating SLAMF6 away from the IS. In the case of loss of synergistic effect, there would be a need to show or state that equal loading of anti-SLAMF6 was performed across the conditions.

Section: SLAMF6 co-immunoprecipitation (co-IP) assay identifies downstream proteins that bridge SLAMF6 and TCR signaling.

The authors propose that plate anti-CD3 with soluble anti-SLAMF6 would deprive or hijack the immune synapse of the components necessary for TCR signalling. This data is weakly supportive or suggestive of such a phenomenon, as some of the soluble SLAMF6 would likely also be engaging within the immune synapse. It would be necessary to show either the TCR or CD3 co-receptor is differentially detected in the two conditions or show a difference in the signalling state of the components (e.g. phosphorylation state). A deeper analysis of the mass spectrometry data may be required, with an expected time frame of 1 to 2 weeks.

4. For Figure 2A, can the authors show whether the TCR or the CD3 co-receptor could be detected within the same assay dataset, as this is an important control to show that SLAMF6 is interacting within or outside the IS ? This would help reinforce and support the authors' statement for this section. Additionally, is the state (e.g. phosphorylation state) of those signalling intermediates showing TCR downstream activation or inhibition? The authors should also indicate the technical replicates or the number of independent experiments performed in Figure 2A.

5. Please rephrase the sentence " A string analysis of the identified SLAMF6 interactome, along with a cartoon diagram of their roles in TCR signaling is shown (Fig. 2B and Fig. 2C)". Perhaps use "schematic" as opposed to "cartoon".

Section: SLAMF6 mediated T cell activation is enhanced when SLAMF6 and CD3 cluster in the immunologic synapse.

The authors show supportive evidence that SLAMF6 and CD3 clustering within the immune synapse would enhance activation. Additional supporting imaging data would further strengthen the weaker notion that soluble anti-SLAMF6 is indeed targeting outside the immune synapse.

6. In Figure 3B, the authors showed that Jurkat T cells and Raji B cells loaded with SEE, had SLAMF6 clustering at the IS

region. Utilising this same imaging model, could the authors test whether the addition of soluble anti-SLAMF6, prevent or reduce SLAMF6 from clustering at the IS, as postulated in Figure 1 and Figure 2.

7. In Figure 3C, the word "sepatated" was used as label, should it be "separated" ?

8. For Figure 3 summary, the authors stated that "...resulted in enhanced T cell activation and established SLAMF6 as an activating coreceptor when localized in proximity to CD3 and an inhibitory co-receptor when separated from the CD3 complex..". However, from the data in Figure 3C-3D, the IL-2 ELISA results, did not show any inhibitory effect (p value is insignificant) in separated beads or antibodies as compared to CD3 alone. Rather it reflected a loss of synergistic activation from "same beads" or "cross-linked antibodies" as compared to segregated condition.

Section: T cell stimulation with anti-CD3/SLAMF6 bispecific antibody enhances T cell activity.

The authors have validated the bi-specific anti-CD3/SLAMF6, and showed that synergistic activation is achieved.

9. Typo in Figure legend 4C " anti-CD3 + ant-SLAMF6 ".

10. The authors stated that " As we had predicted, increased IL-2 secretion was seen following stimulation with anti-CD3/SLAMF6 as compared to either anti-CD3 alone or anti-CD3 in combination with soluble anti-SLAMF6 (Fig. 4C)". To support this statement, please also perform the statistics in Figure 4C, with the bi-specific (CD3/SLAMF6) antibodies against anti-CD3 alone condition.

Section: T cell stimulation with anti-CD45/SLAMF6 bispecific antibody works predominantly in cis to augment T cell activation.

The authors show that the second bi-specific anti-CD45/SLAMF6 gave an unexpected synergistic activation response. It is not clear, whether the bi-specific anti-CD45/SLAMF6 operates within or outside the immune synapse, and may require further supportive data. Additionally, as CD45 and SLAMF6 is also expressed on B cells, it is important to determine the cellular targets as well. The authors should also further elaborate on the aim of the cis/trans experiment.

11. For Figure 5Ci-5Cii, there is residual localisation of SLAMF6 in the IS after the usage of bi-specific CD45/SLAMF6. As alluded by the author in the discussion, it is unsure whether CD45 is truly excluded from the IS or excluded at a specific instance of activation. Would a CD45-GFP (or alternative experiment) be useful in examining whether the bi-specific anti-CD45/SLAMF6 is operating within or outside the IS ? Additionally as CD45 and SLAMF6 is expressed on B cells, the specificity of the cellular target of the bi-specific anti-CD45/SLAMF6 needs to be determined.

12. For Figure 5G, both cis and trans format of stimulating cells, induce an increase of IL-2 as compared with SEE stimulation alone. Does the cis or trans implicitly refer to SLAMF6 operating within or outside the IS. The authors should clarify the aim of this experiment, as it is unclear what the authors are trying to prove or suggest in this experiment.

Section: Co-stimulation of primary mononuclear cells with anti-CD45/SLAMF6 augments T cell activation.

The authors show supportive evidence that the bi-specific anti-CD45/SLAMF6 could induce synergistic activation in human PBMC culture. Minor comment.

13. The authors should taper down on this statement " ..This suggests that, in presence of TCR activation, the bispecific antibody effect on T cell activation is independent of the surrounding immune cell composition ..". The authors examined only two cytokines in this assay, whether other aspects of TCR activation is differential is not known.

Reviewer #1

This paper is focused on mechanisms involved in the regulatory effects of self-binding immune receptor SLAMF6 in T lymphocytes, and offers a spatial explanation for the contradicting effects attributed to this receptor in past reports. Basically, the paper claims that bispecific anti-CD3/SLAMF6 antibodies promote SLAMF6 clustering with CD3 and thus enhanced T cell activation. To show this several experiments repeat a basic structure of T cells/Jurkat cells functional assessment mainly via cytokine secretion. To enhance clustering of CD3 and SLAMF6, the authors use plate-bound aCD3 and aSLAMF6, or beads coated by the antibodies. In addition to cytokine secretion, CFSE is used in Jurkat cells for proliferation, and mass spectrometry to read pull-down protein capture. Then, the bispecific anti-CD3/SLAMF6 antibodies is used to enhance CD3-SF6 proximity.

Here are some comments on data sections:

SLAMF6 mediated T cell activation requires spatial co-localization of SLAMF6 and CD3 along the cell membrane-

Data presented in Figure 1 show primary CD3 T cell activation and compare plate-bound CD3 with SLAMF6-targeting Abs to soluble anti-SLAMF6 Ab. The first enhance cytokine secretion, while the latter reduces it. Data would have been more convincing if the T cells would secrete higher IL-2 or IFN γ under CD3 stimulation (given with 3 μ g/ml, which is high enough), but nevertheless, the data in favor of SF6/CD3 synergy is consistent, with a p value of 0.05, to support the working hypothesis.

Thank you for this comment. These experiments were performed without the addition of CD28. Also, the concentration of the IL-2 is a function of number of cells per volume reaction, as well as T-cell source. We repeated this experiment with 400K primary human T cells per 200 microliter cell volume, and measured IL-2 levels over 96 hour. The results are shown in Figure 1D, with IL-2 levels ranging from 0 to 1500 pg/mL in that time period. IFN- γ levels from human primary T cells range from 0-450 pg/mL under the same conditions.

SLAMF6 co-immunoprecipitation (co-IP) assay identifies downstream proteins that bridge SLAMF6 and TCR signaling.

Network derived of a pull-down experiment that generally show higher peptide scores of SF6-related adaptors and binders with the cluster enhancing stimulation. The immune precipitation shows a non-differential pulldown of essential T cell signaling proteins in both the soluble and plate SLAMF6 conditions. The authors explain this similarity as the reason why soluble SF6 binding is actually weakening the CD3-related stimulation. Any other reference to signaling, including phosphor-signals is not provided. The reference to pull-down as reflecting short-term signals is insufficient, considering the strong background of this assay- and the authors report of only a series of signal proteins while ignoring others.

Thank you for this comment. We performed a deeper analysis of the mass spectrometry data, including functional pathway analysis, now included in Figure 2A, and a protein-protein interaction analysis with kinase mapping (supplement). We did not have access to phospho-data in the IP and added this in the discussion as a limitation of the IP data.

SLAMF6 mediated T cell activation is enhanced when SLAMF6 and CD3 cluster in the immunologic synapse.

This section adds that the need for proximity between SF6 and CD3 is relevant to the immune synapse. The interacting cells are Jurkat- as effectors, and Raji- as APCs. A single synapse's focal microscopy is shown and cytokine release from the Jurkat cells confirms data from figure 1. Jurkat cells vary in their sensitivity to anti-CD3 surface domain binding and for activation, the use of PMA and ionomycin is required.

The experiments are designed to study the localization of co-receptor SLAMF6 with respect to the TCR/CD3 complex on the cell surface membrane. We keep all other experimental conditions identical and repeat all experiments in an attempt to control for variability.

Activation with PMA and Ionomycin bypasses the T cell membrane receptor, and thus would be difficult to utilize in the study of cell surface receptor clustering.

T cell stimulation with anti-CD45/SLAMF6 bispecific antibody works predominantly in cis to augment T cell activation

This construct was design to be a negative control for the previous bispecific Ab, hypothesizing that it would spatially segregate SLAMF6 from the CD3. Given that this hypothesis fell- no reduced reactivity occurred, the authors suggest that it mainly act in cis, and used the reagent in a mixed population of cells.

Thank you for this comment. Indeed, we attempted to differentiate trans vs cis binding in Fig 5G, with an improved explanation of that experiment now included in the main text. In addition, we added new data where we use this antibody on isolated primary human T cells instead of mixed population of cells (Fig. 6D).

Strengths:

- 1- The experiments described yield consistent data, taking into consideration that they scan only a limited repertoire of T cell functions.
- 2- The bispecific anti-CD3/SLAMF6 antibodies are a new idea and validated in cytokine release assays
- 3- The simple yet elegant methodologies that were used to differentiate CD3-SF6 clustering-enhancement versus a non-clustering situation of a soluble antibody.

Thank you for highlighting these strengths.

Weaknesses:

- 1- The experimental design focuses mainly on cytokine secretion with very few- if at all, additional surveys of other aspects of T cell activation.

While cytokine secretion was the main assay used, we also show a proliferation assay in Fig. 1. To address this comment, we added several flow cytometry experiments where we measured CD25 (Fig 1Ci), PD1 (Fig 1Cii) and CD69 (Fig 6Di) as additional markers of activation.

2- Phenotypical markers of activation, cytotoxicity markers, level of maturation are all not included.

We agree with this comment and accordingly added flow cytometry experiments to look at additional phenotypical markers of activation as discussed above (CD25, PD1, CD69). We additionally analyzed the levels of CD4 and CD8 maturation by assessing for CCR7 and CD45RA expression (Fig 1 G). In both of these designs we find a trend for increased T cell activation / maturation in conditions where SLAMF6 and CD3 clustering are encouraged ("plate" condition).

3- Deconstruction of primary CD3 T cells into CD4 and CD8 is not provided.

This is correct. We have added deconstruction into CD4 and CD8 where relevant (T Cell maturation in Fig 1G). On the other hand, where we found the relative trends to be similar between CD4 and CD8 T-Cells, we chose to leave the data for all CD3 T cells for simplicity.

4- The level of expression of SLAMF6 on the APC Raji cells- is not shown, which is a critical element in determining the role of SF6 when there is trans-activation.

Thank you for this comment. That is correct, we did not study SLAMF6 activity on the Raji cells. To overcome the issue of trans activation, we pre-treated Jurkat T cells with the antibodies followed by a wash to remove unbound antibody before co-culturing with the Raji B cells. We also attempted to differentiate trans vs cis binding in Fig 5G, with an improved explanation of that experiment now included in the main text.

5- A single micrograph is shown to support the idea that SF6 clusters in the IS as a general phenomenon

We have added additional imaging of SLAMF6 enrichment and exclusion, from the synapse. (Fig 5C)

6- Some of the data relies on Jurkat cells, in which the CD3/TCR complex is disrupted

Thank you for pointing this out. In response, we have added additional data from primary T-cells to validate the findings in Jurkat T cells. (Fig 1 C, D, E and G and Fig 6 C and D)

7- No data is shown what happens- as in real life- when the TCR is triggered via epitope-MHC conjugates and not by a non-specific anti-CD3-induced receptor cross-binding

We would like to thank the reviewer for this comment; however, we feel that addressing this weakness is beyond the scope of the current work. Our future goal is to mice the murine system where the use of the OTI and OTII system is available.

8- The data on the bifunctional SF6-CD45, which was originally meant to remote SF6 from CD3 and be used as a negative control, is not contributing to the main theme.

We understand this comment and appreciate the feedback. The antibody did not work as expected, but its ability to activate T cells may still have possible therapeutic implications in the future. Since the other

reviewers have asked us to include, and even expand, this data, we did not remove this content from the manuscript.

9- The basic level of almost 0 activation by SEE indicates that there is a co-inhibitory interaction between the Raji and the T cells, otherwise I would expect a basic response of the CD4 component of the T cell to the SEE.

We agree with the reviewer that SEE addition to a RAJI-JKT co-culture is sufficient for T cell activation. These co-cultures were done in 96 well plates with 100-200K JKT T cells. Indeed there is IL-2 release (approximately 200pg/mL) that is not 0, but this IL2 level dwarfs by comparison to the IL-2 released in presence of the bispecific antibody.

To summarize: while the basic hypothesis is interesting and the bi-specific construct data is new, the general picture is based on restricted survey of T cell functions and leave significant vacancies in critical aspects of immune synapse-induced T cell activation

We strongly believe that by adding new data (1) assessment of cell-surface activation markers, (2) measurement of maturation index, (3) analysis of proximal CD3 zeta phosphorylation, (4) kinetics of IL2 production) we were able to enrich the survey of T cell functions as discussed above.

Reviewer #2

Gartshteyn et al. describes their research on the role of SLAMF6 in T cell activation and the regulation of SLAMF6 activity by bispecific antibodies. This is an interesting work that potentially paves way to a novel immunoregulatory drug target molecule. However, there are some major and minor concerns that need to be addressed before the manuscript can be considered for publication.

We would like to thank the reviewer for his/her words, as we strongly believe that these bispecific antibodies could serve the basis of future therapeutic intervention.

Major points

1. The authors used a soluble anti-SLAMF6 antibody for the sequestration of SLAMF6 from IS. Please provide a detailed explanation about how the soluble antibody forces SLAMF6 out of IS, as opposed to uniform distribution (i.e. not concentrated in IS) over the entire cell surface, in which case the result would be similar to an isotype control antibody.

When aSLAMF6 and aCD3 are plate bound, clustering of SLAMF6 and CD3 on the T cell surface is enhanced. On the other hand, aSLAMF6 in solution is physically removed from aCD3 on the plate surface, thus contributing to lack of SLAMF6/CD3 clustering. In cases of an IS (as in a Jurkat-RAJI co-culture), the bispecific antibodies contribute to steric hindrance and reduction of colocalization at the IS. We see this in our microscopy images, but we also know this from a 2014 Nature paper that showed that molecules sized >32nm are excluded from the synapse while those 10-13nm in size (the size of monoclonal antibodies) are reduced in the IS due to steric hindrance. (Cartwright et al. *The immune synapse clears and excludes molecules above a size threshold*. Nature. 2014).

2. A detailed analysis of Fig. 2A is needed. It is not clear what "a non-differential pulldown of essential T cell signaling proteins" means, when there are more than twofold difference between the "plate" and the "soluble" sets for some of the proteins (Src, SAP, ZAP70, etc.). In my opinion this is the most important

molecular-level data supporting the author's claim that SLAMF6 is stimulatory when colocalized in IS but inhibitory when kept away from it.

We agree with the reviewer that this data is strong supporting evidence. Accordingly, we reanalyzed the mass spectrometry data. After excluding the nonspecific proteins associated with empty bead pulldown, we analyzed the remaining proteins in each group by means of a functional pathway analysis. (Fig 2A) Additionally, we performed a protein-protein interaction analysis, identifying relevant kinases that are predicted to be functional downstream of clustered vs. spatially removed SLAMF6 / CD3 activation. (supplement). Thus while pulldown proteins map to the similar pathways in both conditions (T cell receptor signaling pathway), the distal protein-protein interactions and thus kinase activity, propagate along different intracellular signaling cascades.

3. Please provide some comments (maybe in discussion section) about what causes SLAMF6 colocalize in IS during T cell activation.

Thank you for this comment. This has been added to the discussion section as follows:

“It is possible that SLAMF6 and/or TCR ligation results in cytoplasmic assembly of signaling units that result in directed movement of the co-receptor along the cell surface. However, our group has previously shown that while vital for cell activation, the SLAMF6 ectodomain is not necessary to initiate SLAMF6 trafficking to the synapse, suggesting migration signals beyond the ectodomain activation are responsible. [Dragovich et al. SLAMF6 clustering is required to augment T cell activation PLOS ONE 2019]”

4. In page 14, the authors stated that "anti-CD45/SLAMF6 would not be expected to activate resting T-cells in the absence of TCR engagement". Please provide data supporting this claim (i.e. T cells treated by anti-CD45/SLAMF6 alone vs. aCD3 + anti-CD45/SLAMF6).

Thank you for bringing up this important point. In the JURKAT-RAJI co-culture, the addition of the anti-CD45/SLAMF6 had no effect on T-cell activation in the presence of low levels of SEE, suggesting that antigen presentation remains essential, and the bispecific antibody cannot activate resting T-cells. (Fig 4D). We confirmed this finding by taking the reviewer's advice to test primary human T-cell stimulation with aCD45/SLAMF6 alone vs. aCD45/SLAMF6 + aCD3, finding minimal activation following exposure to the bispecific alone. (Fig 6D).

5. Bispecific antibodies (bsAbs) used in this study were made by knobs-into-holes (KiH) technique. Please provide details of bsAb generation in the methods section. Specifically, how did authors solve the problem of the random pairing between heavy and light chains? In antibody engineering perspective, this usually is solved by common light chain approach, or by more elaborate techniques such as CrossMab. Alternatively, a mixture of randomly paired KiH antibodies may be used for the experiments. In any case, it needs to be described in detail how the bsAbs were constructed and produced.

Thank you for the opportunity to clarify our work. The following has been added to the methods:

Bispecific antibodies (bsAbs) used in this study were made by a rational structure-guided approach (ZW1) resulted in a set of substitutions that were reported to lead to over 90% heterodimers with a high thermal stability. The single-chain fragment variable (scFv) of the anti-SLAMF6 Ab and the anti-CD45 Ab were linked to the fragment crystallizable region (Fc) of the two heavy chains included T350V, T366L, K392L, and T394W mutations in the first Fc chain and T350V, L351Y, F405A, and Y407V mutations in the second Fc chain.

Reference: Improving biophysical properties of a bispecific antibody scaffold to aid developability. *MAbs*. 2013 Sep-Oct;5(5):646-54.

6. Could the authors elaborate more on the "two predominant patterns of SLAMF6 expression" (page 7 and Fig. 3B)? Data are provided but explanation is lacking.

Thank you for pointing this out. The explanation has been added:

“Microscopy imaging revealed that there were two predominant patterns of SLAMF6 expression. Specifically, in some cells SLAMF6 expression was evenly distributed across the cell surface membrane (Fig. 3B top row) whereas SLAMF6 enrichment at the site of the IS could be seen in others (Fig. 3B bottom row). The former occurred in resting T cells, as well as in 15% of the cells that were in synapse with another cell; while the remaining 85% of cells in synapse showed SLAMF6 enrichment at the site of the IS.”

Minor points:

1. It is not clear at all what "bivalent anti-CD45-Ig-SLAMF6 antibody" is. Maybe a cartoon showing the structures of bsAbs used in this study would be helpful.

We have added a supplementary cartoon clarifying the structure of the antibodies.

2. bsAb notations are not consistent. Usually it is "anti-CD3/SLAMF6", but in some instances "anti-CD3-SLAMF6" (slash vs. hyphen). Probably the most formal and standard way is using a cross sign (i.e. a CD3 x SLAMF6 bispecific antibody, without "anti-").

Thank you for pointing this out. We have ensured that the bsAb notation is consistent throughout the paper.

3. Full terms for abbreviations should be provided at the first appearance. For example, SEE (staphylococcus enterotoxin E) is used throughout the manuscript, but the full term is not explained until page 16 (materials and methods).

Thank you for noting this. We have made sure to define the abbreviations as they first appear in the paper.

4. V5/GFP tagged SLAMF6 expressing Jurkat T cells -> Jurkat T cells expressing V5/GFP-tagged SLAMF6 (page 5 and 9).

Thank you. We changed the sentences as suggested.

5. Something is missing in the sentence "Indeed, initial recruitment of both CD45 and in early IS formation is required for cell signal propagation" (page 14).

The sentence has been corrected as follows:

“Indeed, initial recruitment of CD45 to the IS is required for cell signal propagation, with expulsion of CD45 due to steric hindrance occurring only subsequently to its initial recruitment.”

6. Page 15: the bsAbs in this study are not "humanized". A humanized antibody refers to an antibody originating from animal (e.g., mouse) and later engineered so that its amino acid sequence closely

resembles that of human antibodies, in an attempt to minimize immunogenicity when administered to human subjects as a therapeutic agent. Therefore "humanized" has nothing to do with "binding to human proteins".

We agree with the reviewer, and this sentence was corrected.

Reviewer #3

Brief description of paper:

The paper describes the effects of SFLAMF6 localisation on CD3 induced TCR activation on T cell / Jurkat T cell or APC-T cell activation. The experimental data shows compelling evidence that SLAMF6 localisation within the immune synapse provides strong synergistic activation. While this phenomenon has been previously published by the authors, the creation and validation of the bi-specific CD3/SLAMF6 antibody does present a substantial application advancement. The unexpected activation response obtained with the application of the second bi-specific CD45/SLAMF6 antibody, while intriguing, would however require further confirmation on its targeting site (outside or within the immune synapse, and the cellular target). On a lesser note, the impact of SLAMF6 localisation outside the immune synapse while novel, is not entirely clear across the experiments. In some instances, SLAMF6 localisation outside the immune synapse represent an inhibitory response (T cell proliferation), in others, it is a loss of synergistic activation (IL-2). This could be due to the difference in sampling time points (24, 48 versus 120 hours), and would require further experiments or clarifications.

We would like to thank this reviewer for his/her time and effort to improve this work. We believe that our bispecific antibodies might have substantial application in the future to treat cancer. We agree that additional experiments with the anti-CD45/SLAMF6 are needed, outside and in the synapse.

To address the concern about sampling timepoints of cytokine measurements, we have performed additional experiments looking at intracellular IL2 by flow after only several hours of activation (Fig 1F) as well as measuring IL2 production over 96 hours (Fig 1D). At all timepoints, we find that clustering of CD3 with SLAMF6 results in enhanced cytokine production, with attenuation of IL2 production seen in the soluble SLAMF6 condition, usually comparable to stimulation with anti-CD3 alone.

Main points of paper:

Section: SLAMF6 mediated T cell activation requires spatial co-localization of SLAMF6 and CD3 along the cell membrane.

The data is supportive of that co-localisation synergistic TCR activation with SLAMF6 using the plate bound anti-CD3/SLAMF6. On a lesser note, the ELISA response from plate bound anti-CD3 with soluble anti-SLAMF6 needs more experimental validation or a re-evaluation on their statement. The expected time frame for experiment would be 1-3 months, depending on reagent availability.

We thank the reviewer for their suggestion, and have added additional experiments better understand T-cell activation patterns that accompany the initial findings in the T cell proliferation assay. Specifically, we added flow cytometric analysis of CD25 and PD1 at 96 hours of activation. We also analyzed IL2 production at different sampling time-points, as suggested by the reviewer. Finally, we assessed T-cell maturation patterns. Please refer to Figure 1 for results of these data.

1. In Figure 1Bii, the authors compared several conditions experimenting with plate bound/soluble anti-

SLAMF6 with plate bound anti-CD3 on T cell proliferation. The presence of soluble anti-SLAMF6 appears to reduce T cell proliferation by plate bound anti-CD3. Did the authors checked if the reduction of T cell proliferation could be caused by unspecific FC-receptor binding of the soluble anti-SFLAMF6? This can be determined by adding the respective specific antibody FC iso-type control for soluble anti-SLAMF6, or alternatively using soluble Fab anti-SFLAMF6.

Thank you for suggesting this. We did perform several experiments early on where we added soluble Fc isotype control to the experiment, but this had no effect/did not dampen the plate CD3 + plate SLAMF6 activation that we have been seeing.

2. In Figure 1C, there was a trend on the inhibitory effect of segregating SLAMF6 on the secretion of IL-2, but it was not significant. Notably, the T cell proliferation assay proceeded for 120hrs, while the ELISA supernatant was harvested within 48hrs. Would an analysis of the cytokines at the 120hrs time point be a better reflection of the observation seen in T cell proliferation?

Thank you for this comment. To evaluate this, we did another experiment to compare cytokine release over a 96 hour period, but found that cytokine production mainly occurred within the first 24 hours and plateaued at 48 hours. (Fig 1D)

However in Figure 1D, the impact of using soluble SLAMF6 with plate anti-CD3 had a synergistic trend compared to anti-CD3 alone for IFN γ secretion. Why is the trend reverse for IFN γ secretion for this condition?

The difference in IFN γ secretion between soluble SLAMF6 and anti-CD3 conditions is non-significant, so we believe that similar to IL2 levels, soluble SLAMF6 dampens the effect of plate SLAMF6 and is similar to anti-CD3 alone.

Additionally for Figure 1C-1D, the figure legends mentioned that technical triplicates were done for each condition, yet it can be clearly seen in the diagram that certain conditions had more than 3 data points. I am not sure if this is a graphical issue or otherwise. If the data points are not equivalent across the conditions, this could have affected the statistics, perhaps the p-value could be properly re-calculated.

The IL2 data has been replaced by an IL-2 ELISA over 96 hours, which eliminates this problem. In 1E, the “control” and “aCD3” conditions have extra technical repeats, but the spread in the values is minimal. For this reason, we chose to not remove original data from the plot.

3. In the summary for Figure 1, the authors need to differentiate between the interpretations that the difference is (a) an inhibitory effect or (b) loss (dampening) of synergistic effect when segregating SLAMF6 away from the IS. In the case of loss of synergistic effect, there would be a need to show or state that equal loading of anti-SLAMF6 was performed across the conditions.

We agree that most of our data suggest a dampening of a synergistic effect when segregating SLAMF6 away from the IS, and have rephrased it as such. Equal loading of anti-SLAMF6 was always ensured across all conditions.

Section: SLAMF6 co-immunoprecipitation (co-IP) assay identifies downstream proteins that bridge SLAMF6 and TCR signaling.

The authors propose that plate anti-CD3 with soluble anti-SLAMF6 would deprive or hijack the immune synapse of the components necessary for TCR signaling. This data is weakly supportive or suggestive of such a phenomenon, as some of the soluble SLAMF6 would likely also be engaging within the immune

synapse. It would be necessary to show either the TCR or CD3 co-receptor is differentially detected in the two conditions or show a difference in the signalling state of the components (e.g. phosphorylation state). A deeper analysis of the mass spectrometry data may be required, with an expected time frame of 1 to 2 weeks.

Thank you for this comment. We performed a deeper analysis of the mass spectrometry data, including functional pathway analysis, now included in Figure 2A, and a protein-protein interaction analysis with kinase mapping (supplement). We were unable to IP the TCR and added this in the discussion as a limitation of the IP data.

4. For Figure 2A, can the authors show whether the TCR or the CD3 co-receptor could be detected within the same assay dataset, as this is an important control to show that SLAMF6 is interacting within or outside the IS ? This would help reinforce and support the authors' statement for this section. Additionally, is the state (e.g. phosphorylation state) of those signalling intermediates showing TCR downstream activation or inhibition? The authors should also indicate the technical replicates or the number of independent experiments performed in Figure 2A.

We identified the CD3 delta subunit in the SLAMF6 protein interactome. While the peptide count was too small to make any conclusions about differential expression between the two conditions, it does support that SLAMF6 is interacting with the CD3/TCR complex.

The IP experiment was done in three repeats. This is now included in the legend for Figure 2.

5. Please rephrase the sentence " A string analysis of the identified SLAMF6 interactome, along with a cartoon diagram of their roles in TCR signaling is shown (Fig. 2B and Fig. 2C)". Perhaps use "schematic" as opposed to "cartoon".

This sentence has been removed.

Section: SLAMF6 mediated T cell activation is enhanced when SLAMF6 and CD3 cluster in the immunologic synapse.

6. The authors show supportive evidence that SLAMF6 and CD3 clustering within the immune synapse would enhance activation. Additional supporting imaging data would further strengthen the weaker notion that soluble anti-SLAMF6 is indeed targeting outside the immune synapse. In Figure 3B, the authors showed that Jurkat T cells and Raji B cells loaded with SEE, had SLAMF6 clustering at the IS region. Utilising this same imaging model, could the authors test whether the addition of soluble anti-SFLAM6F, prevent or reduce SFLAM6F from clustering at the IS, as postulated in Figure 1 and Figure 2.

Please allow us to explain why we don't think this would be possible. We use the plate bound anti-CD3 and plate bound anti-SLAMF6 to mimic the clustering that would occur in the setting of an IS in a co-culture. We use soluble aSLAMF6 to disperse the SLAMF6 away from the aCD3 coated plate surface. In either situation, there is no synapse – only manipulation of CD3 and SLAMF6 molecules. As a result we are not able to apply our microscopy set up to image the plate vs. soluble experimental setups from Figure 1 and Figure 2.

7. In Figure 3C, the word "sepatated" was used as label, should it be "separated" ?

Thank you, this has been corrected.

8. For Figure 3 summary, the authors stated that "...resulted in enhanced T cell activation and established SLAMF6 as an activating coreceptor when localized in proximity to CD3 and an inhibitory co-receptor when separated from the CD3 complex." However, from the data in Figure 3C-3D, the IL-2 ELISA results, did not show any inhibitory effect (p value is insignificant) in separated beads or antibodies as compared to CD3 alone. Rather it reflected a loss of synergistic activation from "same beads" or "cross-linked antibodies" as compared to segregated condition.

Thank you for pointing this out. We agree that the cytokine data suggest this is more of a loss of synergistic activation as compared to inhibition. We have changed this in the main text to read as follows:

"Physically promoting SLAMF6 clustering with CD3 at time of stimulation, using either a system of conjugated beads or cross-linkers, resulted in enhanced T cell activation and established SLAMF6 as an activating co-receptor when localized in proximity to CD3. On the other hand, T cell activation was dampened when SLAMF6 was spatially removed from the CD3 complex, suggesting loss of synergistic signaling."

Section: T cell stimulation with anti-CD3/SLAMF6 bispecific antibody enhances T cell activity.

The authors have validated the bi-specific anti-CD3/SLAMF6, and showed that synergistic activation is achieved.

9. Typo in Figure legend 4C " anti-CD3 + ant-SLAMF6 ".

Thank you. We apologize for this typo and have corrected it.

10. The authors stated that " As we had predicted, increased IL-2 secretion was seen following stimulation with anti-CD3/SLAMF6 as compared to either anti-CD3 alone or anti-CD3 in combination with soluble anti-SLAMF6 (Fig. 4C)". To support this statement, please also perform the statistics in Figure 4C, with the bi-specific (CD3/SLAMF6) antibodies against anti-CD3 alone condition.

Thank you. This analysis is now included.

Section: T cell stimulation with anti-CD45/SLAMF6 bispecific antibody works predominantly in cis to augment T cell activation.

The authors show that the second bi-specific anti-CD45/SLAMF6 gave an unexpected synergistic activation response. It is not clear, whether the bi-specific anti-CD45/SLAMF6 operates within or outside the immune synapse, and may require further supportive data.

We have added additional microscopy data (see question #11 below) to confirm that treatment of T-cells with anti-CD45/SLAMF6 excludes both CD45 and SLAMF6 from clustering within the IS with B-cells. We acknowledge that we do not fully understand how anti-CD45/SLAMF6 operates in cis to enhance T-cell activity, and this will be the topic of future work.

Additionally, as CD45 and SLAMF6 is also expressed on B cells, it is important to determine the cellular targets as well.

Thank you for this comment. To answer this question, we treated primary human T-cells with anti-CD3 +/- anti-CD45/SLAMF6. No APCs were added to this assay. We observed increased CD69 expression and increased IL2 and IFN γ secretion in response to anti-CD45/SLAMF6 confirming that T cells were the

cellular targets of the bispecific antibody. Additionally, T cells treated with anti-CD45/SLAMF6 in the co-culture experiments were washed (for removal of unbound anti-CD45/SLAMF6) before being incubated with RAJI B Cells, but this did not change the microscopy results.

The authors should also further elaborate on the aim of the cis/trans experiment.

Thank you for giving us the opportunity to explain the aim of this experiment. The text has been edited:

“In an attempt to better understand how anti-CD45/SLAMF6 resulted in T cell activation despite decreased enrichment of SLAMF6 in the IS, we sought to investigate whether the antibody binds in trans or in cis. Specifically, the aim of this experiment was to compare T-cell activation when the anti-CD45/SLAMF6 was added directly into a T – B cell co-culture (trans binding) as compared to adding the anti-CD45/SLAMF6 to T-cells (and washing off unbound antibody) prior to co-culturing with B cells (cis binding).”

11. For Figure 5Ci-5Cii, there is residual localisation of SLAMF6 in the IS after the usage of bi-specific CD45/SLAMF6. As alluded by the author in the discussion, it is unsure whether CD45 is truly excluded from the IS or excluded at a specific instance of activation. Would a CD45-GFP (or alternative experiment) be useful in examining whether the bi-specific anti-CD45/SLAMF6 is operating within or outside the IS ?

Thank you for this suggestion to improve our work. We have generated OFP-CD45 and GFP-SLAMF6 T cells and performed the suggested experiments. Please see the results in Fig 5C. Briefly, we were able to visualize CD45 exclusion from, and SLAMF6 enhancement in, the IS under normal condition. Treatment of T cells with anti-CD45/SLAMF6 did not change CD45 exclusion from the IS, but did reduce SLAMF6 enhancement in the synapse. We could visualize SLAMF6 localization overlapping with CD45 following treatment with the anti-CD45/SLAMF6.

Additionally as CD45 and SLAMF6 is expressed on B cells, the specificity of the cellular target of the bi-specific anti-CD45/SLAMF6 needs to be determined.

The T cells were washed following pre-treatment with anti-CD45/SLAMF6 (to wash off unbound antibody) and before the co-culture, thus minimizing interaction of free bispecific antibody with B cells.

12. For Figure 5G, both cis and trans format of stimulating cells, induce an increase of IL-2 as compared with SEE stimulation alone. Does the cis or trans implicitly refer to SLAMF6 operating within or outside the IS. The authors should clarify the aim of this experiment, as it is unclear what the authors are trying to prove or suggest in this experiment.

Thank you for the opportunity to clarify the aim and findings of this experiment. This has been amended in the text.

Section: Co-stimulation of primary mononuclear cells with anti-CD45/SLAMF6 augments T cell activation.

The authors show supportive evidence that the bi-specific anti-CD45/SLAMF6 could induce synergistic activation in human PBMC culture. Minor comment.

13. The authors should taper down on this statement “. This suggests that, in presence of TCR activation, the bispecific antibody effect on T cell activation is independent of the surrounding immune cell

composition ..". The authors examined only two cytokines in this assay, whether other aspects of TCR activation is differential is not known.

We agree with the reviewer and revised the text as follows:

"This suggests that, in presence of TCR activation, the bispecific antibody has an excitatory effect on T cell activation."

November 22, 2022

RE: Life Science Alliance Manuscript #LSA-2022-01533R

Dr. Yevgeniya Gartshteyn
Columbia University Medical Center
Medicine
630 West 168th Street
New York, NY 10032

Dear Dr. Gartshteyn,

Thank you for submitting your revised manuscript entitled "SLAMF6 compartmentalization enhances T cell functions.". We would be happy to publish your paper in Life Science Alliance pending final revisions necessary to meet our formatting guidelines.

- please address the final Reviewer #3's minor comments
- please add an Abstract to our system
- please add the Twitter handle of your host institute/organization as well as your own or/and one of the authors in our system
- please use the [10 author names, et al.] format in your references (i.e. limit the author names to the first 10)
- please add figure callouts for your supplementary figures to the main manuscript text
- please upload your table files as an editable doc or excel file or make sure that it's included in the doc file of your manuscript
- please add a separate ethics statement

Figure check:

- your figure panels are in the format Ai, Aii etc. Please update your figure panel in the format A, B, C, D, E, F etc.

A. FINAL FILES:

B. MANUSCRIPT ORGANIZATION AND FORMATTING:

Sincerely,

Reviewer #2 (Comments to the Authors (Required)):

In the revised manuscript, the authors addressed major and minor points raised in the first round of review process. I believe that the manuscript is now fit for the publication in LSA.

Reviewer #3 (Comments to the Authors (Required)):

The paper presents findings to support that SLAMF6 localisation within the immune synapse provides strong synergistic activation, in particular, with the usage of bi-specific antibodies CD3/SLAMF6 and CD45/SLAMF6. The authors have made substantial changes to further support these claims, in terms of experimental, analytical and statistical improvements. Furthermore, with the modification of text, there is better clarity to the rationality of certain experiments.

Minor comments:

1. The sentence reads "A cartoon diagram of the identified SLAMF6 interactome in TCR signaling is shown (Fig. 2C)". It may be appropriate to substitute the word "cartoon".
2. Supplement 4 illustration of the antibodies may require proper cropping, as there is a portion of the image on the left bottom corner.

November 29, 2022

RE: Life Science Alliance Manuscript #LSA-2022-01533RR

Dr. Yevgeniya Gartshteyn
Columbia University Medical Center
Medicine
630 West 168th Street
New York, NY 10032

Dear Dr. Gartshteyn,

Thank you for submitting your Research Article entitled "SLAMF6 compartmentalization enhances T cell functions.". It is a pleasure to let you know that your manuscript is now accepted for publication in Life Science Alliance. Congratulations on this interesting work.

DISTRIBUTION OF MATERIALS:

Again, congratulations on a very nice paper. I hope you found the review process to be constructive and are pleased with how the manuscript was handled editorially. We look forward to future exciting submissions from your lab.

Sincerely,
